# Meiotic divisions and round spermatid formation do not require centriole duplication in mice

Marnie W. Skinner[1,2,3], Paula B. Nhan[2], Carter J. Simington[1], Philip W. Jordan[1,2,4]*

**1** Department of Biochemistry and Molecular Biology, Johns Hopkins University Bloomberg School of Public Health, Baltimore, Maryland, United States of America, **2** Department of Biochemistry and Molecular Biology, Uniformed Services University of the Health Sciences, Bethesda, Maryland, United States of America, **3** The Henry M. Jackson Foundation for the Advancement of Military Medicine, Bethesda, Maryland, United States of America, **4** School of Biomedicine, The University of Adelaide, Adelaide, Australia

* philip.jordan@usuhs.edu

## Abstract

Centrosomes, composed of centrioles and pericentriolar matrix proteins, are traditionally viewed as essential microtubule-organizing centers (MTOCs) that facilitate bipolar spindle formation and chromosome segregation during spermatogenesis. In this study, we investigated the role of centrioles in male germ cell development by using a murine conditional knockout (cKO) of *Sas4*, a critical component of centriole biogenesis. We found that while centriole duplication was impaired in *Sas4* cKO spermatocytes, these cells were still capable of progressing through meiosis I and II. Chromosome segregation was able to proceed through the formation of a non-centrosomal MTOC, indicating that centrioles are not required for meiotic divisions. However, spermatids that inherited fewer than two centrioles exhibited severe defects in spermiogenesis, including improper manchette formation, constricted perinuclear rings, disrupted acrosome morphology, and failure to form flagella. Consequently, *Sas4* cKO males were infertile due to the absence of functional spermatozoa. Our findings demonstrate that while centrioles are dispensable for meiosis in male germ cells, they are essential for spermiogenesis and sperm maturation. This work provides key insights into the role of centrosomes in male fertility and may have implications for understanding certain conditions of male infertility associated with centriole defects.

## Author summary

Sperm cells, or sperm, develop through a complex process that includes cell division (meiosis) and a remodeling phase (spermiogenesis), where round cells transform into mature sperm with tails for swimming. In most cells, structures called centrioles help organize the cell's skeleton and assist in proper division.

**Data availability statement:** All data underlying the findings presented in this manuscript are available within the uploaded supplementary information provided.

**Funding:** This work was funded by NIGMS grant to PWJ (R01GM11755), NICHD grant to PWJ (R01HD114180), and training grant fellowships (NCI, NIH; T32CA009110; NICHD, NIH; F31HD111265) to MWS. Partial salary for MWS was covered by F31HD111265 and R01GM11755. Partial salary for PWJ when at JHU was covered by R01GM11755. The funders had no role in the study design, data collection and analysis, decision to publish, or preparation of the manuscript.

**Competing interests:** I have read the journal's policy and the authors of this manuscript have the following competing interests: PWJ is on the scientific advisory board of Gameto, Inc. All other authors have no competing interests to disclose.

However, their role in sperm formation has been unclear. In this study, we genetically modified mice to remove a key protein required for centriole duplication in sperm. Surprisingly, we found that sperm precursor cells could still divide properly without centrioles, using alternative structures to separate their genetic material. However, when these cells reached the remodeling phase, they failed to form normal shaped sperm and could not grow tails, which ultimately led to male infertility. This research reveals that centrioles are not necessary for sperm precursor division but are crucial for the final stages of sperm development. These findings may help explain some cases of human male infertility and provide insights into reproductive health.

## Introduction

Ciliopathies are genetic disorders that result in defects in cilia structure and function [1]. These defects can affect all cell types that rely on either non-motile or motile cilia [1]. The primary function of non-motile cilia is to transduce cellular signals from their environment through signaling pathways such as wingless (Wnt), hedgehog (Hh), and G-protein coupled receptors (GPCRs) [2]. Conversely, motile cilia are involved in cellular movement or the movement of extracellular materials [1]. Cilia are found in the cells of the bronchial epithelium, oviductal epithelium, ependymal cells lining the brain ventricles, and as the flagella of sperm [2]. Sperm flagella are highly specialized cilia structures with unique features that are important for completing fertilization. Specifically, the flagella axoneme differs from other motile cilia in that it directly interacts with the cytosol and mitochondria. Furthermore, the sperm head and flagella are connected via a specialized head-tail coupling apparatus (HTCA) [3,4].

Motile ciliopathies predominantly result in chronic bronchitis, or the inability to clear lung mucus [1]. While observed less often, reduced cerebrospinal fluid movement in the brain ventricles is also reported, leading to increases in headache or hydrocephalus propensity [1]. Motile ciliopathies can also result in fertility defects that include reduced efficiency of oocyte movement through the oviduct and male infertility because of abnormalities during flagella formation that compromise sperm function [1]. Because cilia are critical to multiple cell types, ciliopathies often cause multiple organ defects [5–10]. Examples include conditions such as Bardet-Biedl, Kartagener, and Usher syndromes, which are ciliopathies that present initially as retinal dystrophy, renal dysfunction, or defects in the epithelial cells of the respiratory tract [5–10]. However, male individuals with these syndromes also experience testicular defects such as hypogonadism, which can lead to infertility [6–10]. These symptoms are likely because of defects in the primary structure that contributes to cilia formation, the centrosome.

Centrosomes are membrane-less organelles that consist of two cylindrical microtubule (MT) structures, or centrioles, surrounded by a proteinaceous structure known as the pericentriolar matrix (PCM). During centriole assembly, polo-like

kinase 4 (PLK4) triggers centriole duplication through the phosphorylation of STIL (SCL/TAL-interrupting locus protein; [11–14]). STIL interacts with and promotes the assembly of SAS6 (spindle assembly abnormal protein 6) into a ninefold symmetrical cartwheel-like structure at the proximal end of the centriole [15]. Following cartwheel assembly, the structural centriole protein SAS4 (also known as CENPJ or CPAP) is recruited to the centriole and necessary for the addition of MTs onto the structure [16,17]. The MTs are assembled in triplicate around the cartwheel base with the same ninefold radial symmetry ([11,12,14]). SAS4 controls the rate at which MTs are added to the base structure, and studies overexpressing SAS4 demonstrate excessive centriole MT length [18,19]. The distal end of a centriole central barrel structure is comprised of centrin proteins (CETN1, CETN2, CETN3, and CETN4) and is capped off by the centriolar coiled-coil protein, CP110. [11,12,14]. Removal of CP110 is necessary for the extension of the axoneme during ciliogenesis [20,21]. Centrioles also undergo maturation through the acquisition of distal and subdistal appendages, which are necessary for ciliogenesis and the coordinated recruitment of PCM components to the centrosome, respectively [22,23]. PCM proteins are assembled at the centrosome in a coordinated manner beginning with innermost proteins that include CEP192 and CDK5RAP2, followed by outer proteins such as NEDD1 and γ-tubulin, and ubiquitously localized proteins like that of pericentrin (PCNT; [11,12,14]). PCM proteins play important roles in centriole stabilization and MT nucleation [14].

One of the major roles of the centrosome is serving as the microtubule organizing center (MTOC) during cellular division. In order to act as the MTOC in mitotically dividing cells, centrioles must duplicate during early S-phase to ensure each spindle pole and resulting cell after division harbors two centrioles [24,25]. However, during male meiosis, cells undergo chromosome segregation twice without an intervening S-phase to generate haploid gametes [26,27]. Therefore, to ensure each spindle pole and the resulting haploid spermatids harbor two centrioles, centriole duplication must happen twice during spermatogenesis. The first centriole duplication occurs at the beginning of meiosis, during early prophase (late leptotene stage), and the second centriole duplication takes place after the first meiotic division, during interkinesis [26,27]. After meiosis, the resulting round spermatids each harbor one centrosome containing one mature and one immature centriole [26,27].

The second major role of the centrosome is during a process known as spermiogenesis, where they contribute to spermatid remodeling and flagella formation, which will ultimately result in the formation of mature spermatozoa [24,28]. Spermatozoa consist of two distinct regions, the head and tail. [3,29]. As spermatid remodeling initiates, the centrioles undergo nuclear attachment mediated by the HTCA, designating what will become the neck region of the flagella at the distal end of the head [3,4,29]. The immature or proximal centriole transitions into a structure known as the centriole adjunct and helps facilitate the development of the neck region and orientation of protein trafficking structures [ 3,29]. The mature or distal centriole serves as the basal body for MT extension when forming the flagella axoneme. Axoneme MTs are remodeled from the triplicate arrangement within the centriole, to doublet arrangements, and gain a central pair of MTs. Major remodeling of centriole and PCM proteins also occurs to form flagella structures. Centriole components such as CETN1 and CETN2 are utilized to form rods within the distal centriole [3,29]. PCM components contribute to the capitulum and striated columns located in the neck region [3,29]. These centrosome components are critical for maintaining the head-tail attachment, as exemplified by the fact that mutation of centrosomal genes *Cep112*, *Cep250*, *Cntrob*, and *Odf1*, lead to acephalic spermatozoa (headless sperm) and non-obstructive azoospermia [30–35]. In the head of the spermatid, nuclear remodeling and acrosome biogenesis occur during spermiogenesis [28]. The acrosome cap, consisting of the Golgi apparatus, develops to cover the proximal end of the head [36]. Following this is the elongation stage of spermiogenesis, which consists of a nucleosome-to-protamine exchange. This process begins with the displacement of histones by transition proteins (TNP1 and TNP2), which are subsequently replaced by protamines (PRM1 and PRM2; [37,38]). This transition is crucial for chromatin compaction and nuclear remodeling that is facilitated by the transient MT based protein trafficking structure known as the manchette [39]. Manchette MTs nucleate from the centriole adjunct and extends upwards towards the perinuclear ring [40,41]. The perinuclear ring is a structure that temporarily forms in elongating spermatids at the

hemisphere of the spermatid head that helps stabilize manchette MT plus ends before being disassembled near the end of spermiogenesis [39–42].

In this report, we used a murine conditional knockout (cKO) of *Sas4* to model male infertility caused by a defect in a critical structural component of the centriole. We determined that *Sas4* cKO leads to centriole duplication failure during spermatogenesis. Nevertheless, these spermatocytes still progress through meiosis I and II to form haploid round spermatids harboring only one or zero centrioles. However, defects during spermiogenesis arise in spermatids that do not inherit two centrioles leading to perturbations in the assembly and disassembly of protein trafficking structures, abnormal acrosome and DNA morphology, and the absence of flagella. Consequently, functional spermatozoa fail to develop, resulting in male infertility.

## Results

### *Depletion of SAS4 in spermatocytes leads to male infertility*

SAS4 is an essential structural component of the centriole [43–45]. Since a whole-body deletion of *Sas4* results in embryonic lethality, we utilized a conditional mouse model to study the effect of depleting the protein in spermatocytes [46]. We accomplished this by using a cKO allele for *Sas4* where the fifth exon of the *Sas4* gene was flanked by loxP sites (floxed; [47]; see "Materials and methods"). We utilized a Cre recombinase driven by a *Spo11* promoter (*Spo11-Cre* recombinase). *Spo11-Cre* recombinase excised the floxed exon in the *Sas4* gene at 9 days post-partum (dpp) during the preleptotene/leptotene stage of meiosis I [48–50]. Mice homozygous for the *Sas4* flox allele and hemizygous for the *Spo11-Cre* transgene (*Sas4* flox/flox, *Spo11-Cre* tg/0) were referred to as *Sas4* cKO (Fig 1A and 1B and "Materials and methods").

To test the fertility status, three *Sas4* cKO males were each bred to two fertile females, but no litters were observed after at least 8 weeks. This result implied that *Sas4* cKO males were infertile. In contrast, *Sas4* cKO females produced live litters when bred with fertile males, which indicated that SAS4 is not required for female meiosis. This is consistent with previous work, which found that other centriole components are not required for oogenesis and female fertility [51,52]. This is likely because centrioles are eliminated in oocytes, and instead, acentriolar MTOCs (aMTOCs) mediate chromosome segregation following meiotic resumption [53].

Initial assessments of testis to body weight ratios of control and *Sas4* cKO mice revealed no significant difference in testes size (Fig 1C). We further investigated the cause of *Sas4* cKO male infertility by assessing histological sections of testes and epididymides. Comparison of the control and *Sas4* cKO testes cross-sections stained with hematoxylin and eosin (H&E) demonstrated that *Sas4* cKO testes had significantly larger lumen to seminiferous tubule diameter ratios than control testes (0.42 vs 0.35, respectively; Fig 1D and 1E). The large lumen diameter was due to the *Sas4* cKO elongated spermatids not harboring flagella (Fig 1D). Assessment of H&E-stained epididymis cross-sections revealed that the *Sas4* cKO epididymides harbored very few spermatozoa compared to the controls (Fig 1F). Furthermore, epidydimal sperm counts of *Sas4* cKO mice were approximately 100 times less than their littermate controls (Fig 1G). While these results indicate SAS4 depletion prevents normal sperm production, closer assessment of spermatogenesis was required to identify the primary cause of the infertility defect.

### Spermatocytes successfully form bipolar spindles and complete chromosome segregation despite centriole duplication failure

To identify how SAS4 depletion was perturbing sperm production, we first looked for defects during meiotic prophase progression. Previous reports demonstrated that in addition to impaired centriole duplication during spermatogenesis, a gene-trap mutation of the pericentriolar component *Cep63* resulted in defects in homologous recombination and chromosome synapsis, leading to subsequent germ cell apoptosis during meiotic prophase [54]. Therefore, we utilized an antibody against phospho-histone H2A.X (γH2AX) to assess DNA damage and repair during meiotic

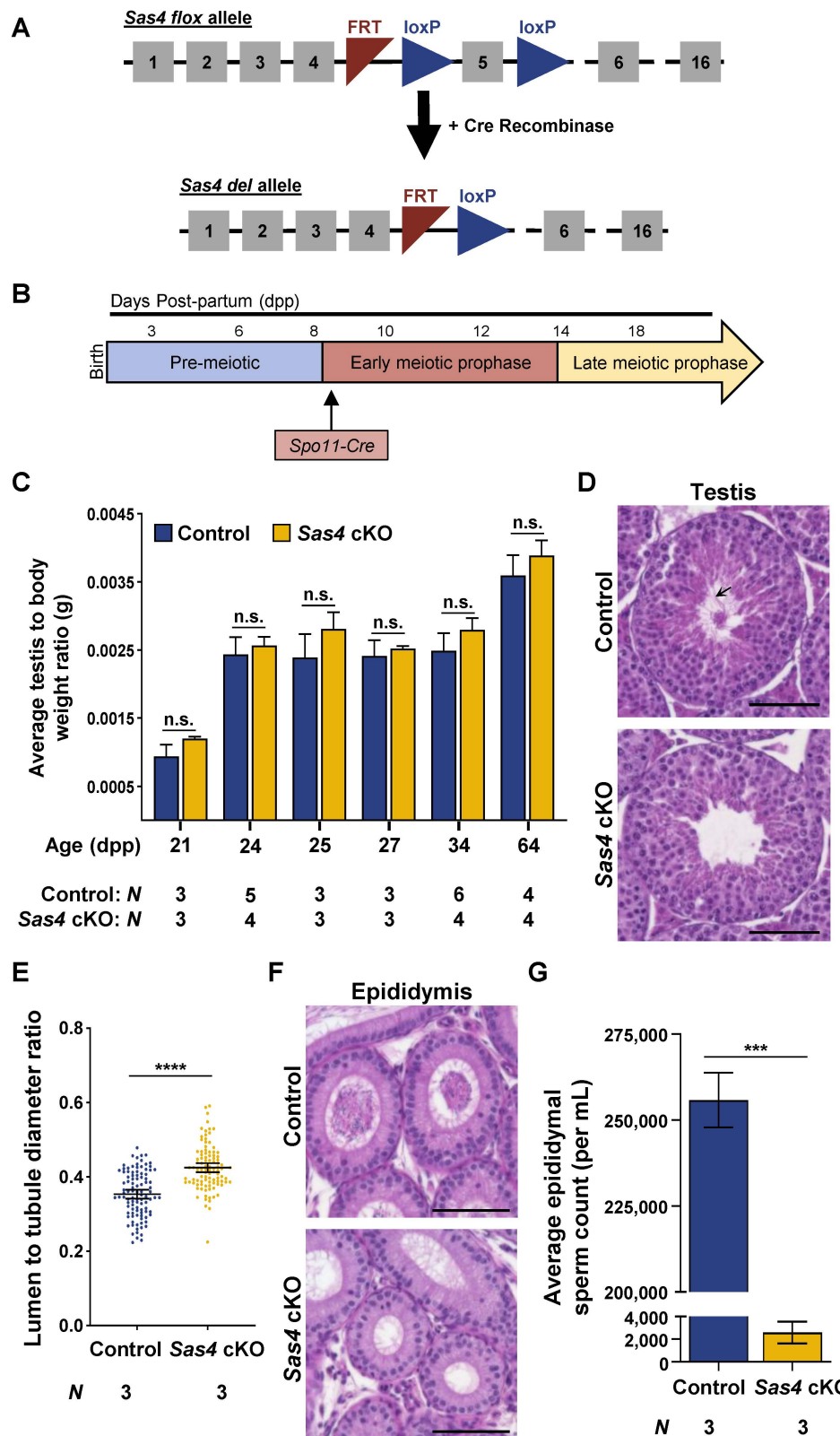

**Fig 1. Depletion of SAS4 in spermatocytes leads to male infertility. A.** Diagram of the *Sas4* cKO allele before and after Cre-mediated recombination. The *Sas4* flox allele harbors two loxP sites (blue triangle) flanking exon 5 (grey box). Excision of exon 5 by Cre recombinase results in the *Sas4*

deletion allele (*Sas4* del allele). **B.** The *Spo11* promoter used to drive Cre recombinase expression in the *Sas4* cKO mouse model is expressed in early meiotic prophase at ~9 dpp. **C.** Quantification of the average testis to body weight ratio of control and *Sas4* cKO mice. Measurements were performed using ≥ 3 mice for 21, 24, 25, 27, 34, and 64 dpp. P values for 21, 24, 25, 27, 34, and 64 dpp were 0.1161, 0.3504, 0.1699, 0.5200, 0.0536, and 0.1143, respectively. Error bars show mean ± SEM. P values obtained from two-tailed Student's t-test. n.s. (not significant). **D.** H&E staining of 5 μm thick testis sections of 44 dpp control and *Sas4* cKO mice. Black arrow indicates flagella located in the lumen of the seminiferous tubules. Scale bar = 80 μm. **E.** Quantification of the average lumen diameter to tubule diameter ratio in control and *Sas4* cKO mice. The average tubule diameter was 326.75 μm and 434.05 μm in control and *Sas4* cKO mice, respectively. The average tubule lumen diameter was 116.10 μm and 185.38 μm in control and *Sas4* cKO mice, respectively. Quantification was performed in three biological replicates with ≥33 tubules quantified per replicate. The total number of tubules measured for the control and *Sas4* cKO was 100 each. P value = < 0.0001. Error bars show mean ± 95% CI. P values obtained from two-tailed Student's t-test. ****P < 0.0001. **F.** H&E staining of 5 μm thick epididymides sections of 226 dpp control and *Sas4* cKO mice. Scale bar = 80 μm. **G.** Epididymal sperm count in control and *Sas4* cKO mice, respectively. The average sperm count was 255,833 sperm/mL and 2,583 sperm/mL in control and *Sas4* cKO epididymides, respectively. Quantification was performed using three biological replicates. P value = 0.0008. Error bars show mean ± SEM. P values obtained from two-tailed Student's t-test. n.s. ***P < 0.001.

prophase. Localization of γH2AX in control spermatocytes presents as widespread staining across most of the chromatin during leptotene and early zygotene stage spermatocytes [55,56]. In mid-late zygonema, double-strand breaks are repaired, and the γH2AX signal is reduced [55,56]. By pachynema and continuing into diplonema, the majority of DNA repair has been completed, and γH2AX localizes primarily to the X-Y chromosome pair in male gametes [55,56]. There was no observable difference in DNA damage and repair or sex body formation between the *Sas4* cKO and control prophase I spermatocytes, based on γH2AX localization (Fig 2A). We did not detect any defects in synapsis and desynapsis based on the localization of the lateral element protein of the synaptonemal complex, SYCP3 (Fig 2A). We also scored no difference in crossover recombination site formation during pachynema when observing control and *Sas4* cKO spermatocytes immunolabeled against MLH1, which marks the majority of crossover recombination sites (Fig 2B and 2C; [57]). These observations are consistent with recent findings from analysis of the *Plk4* cKO, which also demonstrated no defects in meiotic recombination or chromosome synapsis/desynapsis during spermatogenesis [51].

Thus, *Plk4* cKO and *Sas4* cKO spermatocytes do not exhibit defects in homologous recombination or chromosome synapsis and desynapsis, which contrasts with what was reported for the *Cep63* gene-trapped allele. This difference may be due to the *Cep63* mutation being present in all cells and tissues of the mouse, instead of specifically within the germ cells, as is the case for the *Plk4* cKO and *Sas4* cKO [51,54]. *Cep63* mutation also causes microcephaly, which may affect the hypothalamus-pituitary-gonadal axis, and other cell types that support gametogenesis could also be impacted [58]. Alternatively, CEP63 may have additional functions beyond its role at the centrosome, whereas PLK4 and SAS4 are exclusively required for centriole biogenesis [51,54,59,60].

Based on the previously reported role of SAS4 being essential for centriole duplication, we next assessed centriole biogenesis during prophase I of spermatogenesis [45,51]. To determine the centriole number in spermatocytes, we looked for the presence of the centrin centriole proteins using either a CETN3 antibody or observing the expression of CETN2-GFP [61]. We first observed centriole numbers in early prophase I to determine if the centriole duplication during leptonema was successful in *Sas4* cKO primary spermatocytes. As previously reported, control spermatocytes entered meiosis with 2 centrioles (Fig 3A and 3B; [26]). By zygonema, they had undergone centriole duplication and harbored 4 centrioles (Fig 3A and 3B). By metaphase I, these four centrioles underwent separation, and each spindle pole contained 2 centrioles (Fig 3C and 3D). During interkinesis, the second centriole duplication event occurs, and subsequent control metaphase II spermatocytes exhibited successful duplication and separation with each spindle pole harboring 2 centrioles (Fig 3D and 3E).

*Sas4* cKO primary spermatocytes entered meiosis with 2 centrioles, and by zygonema, 72% still only harbored 2 centrioles (Fig 3A, 3B and 3F). While the majority of spermatocytes failed to undergo centriole duplication, a proportion of spermatocytes successfully duplicated their centrioles. This implies that there was either inefficient Cre-recombinase-mediated

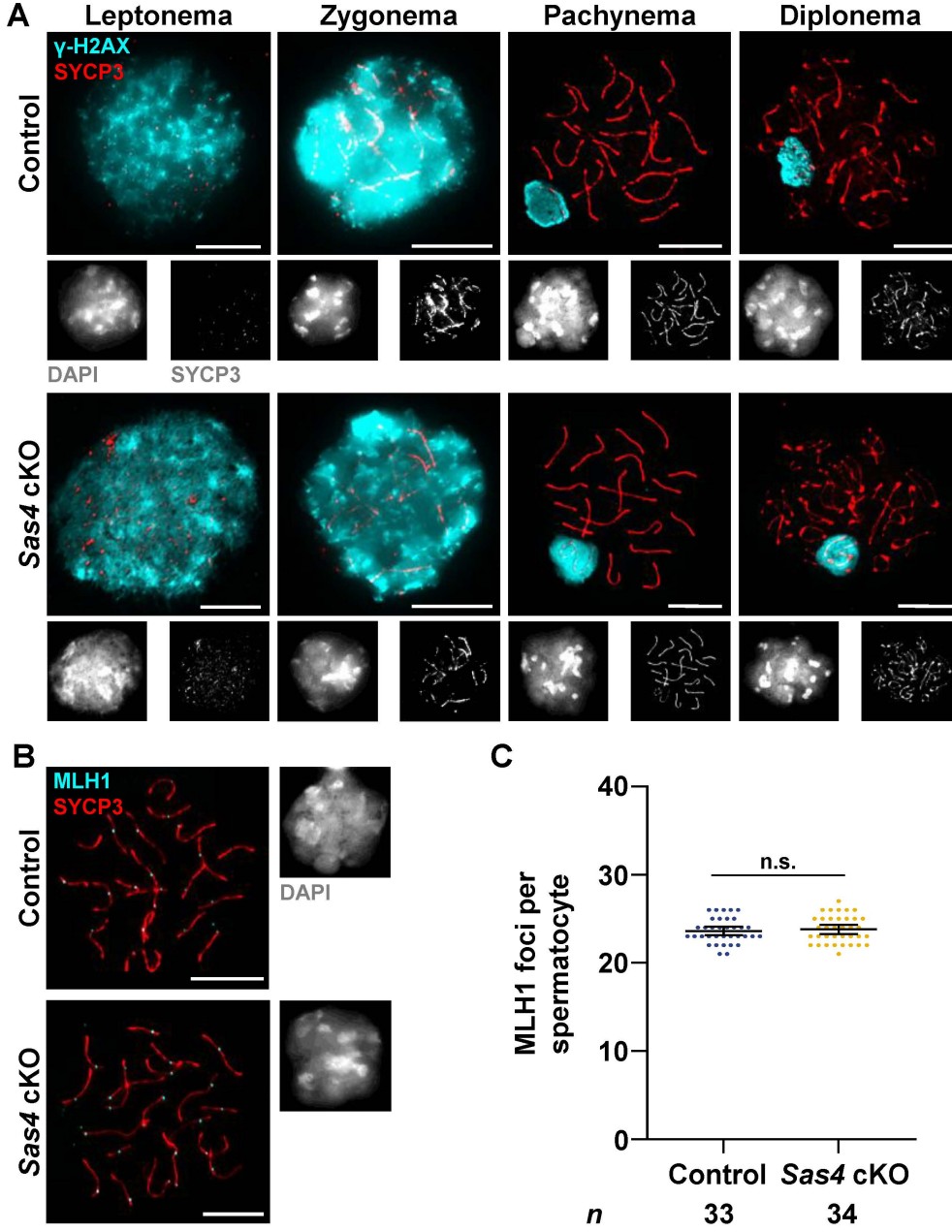

**Fig 2. Characterization of meiotic prophase progression and crossover formation. A.** Prophase I control and *Sas4* cKO spermatocytes from 14 and 20 dpp mice were immunolabeled against γ-H2AX (cyan), SYCP3 (red and grey outset below corresponding image), and stained with DAPI (grey outset below corresponding image). Scale bars = 10 µm. **B.** Representative images of mid-prophase I spermatocytes in 20 dpp control and *Sas4* cKO mice immunolabeled against SYCP3 (red), MLH1 (cyan), and stained with DAPI (grey outset to the right of corresponding image). Scale bars = 10 µm. **C.** Quantification of MLH1 foci observed along SYCP3 stretches during pachynema in both control and *Sas4* cKO spermatocytes. The total number of cells quantified for control and *Sas4* cKO mice were 33 and 34, respectively. P value = 0.7069. Error bars show mean ± 95% CI. P values obtained from two-tailed Student's t-test. n.s. (not significant).

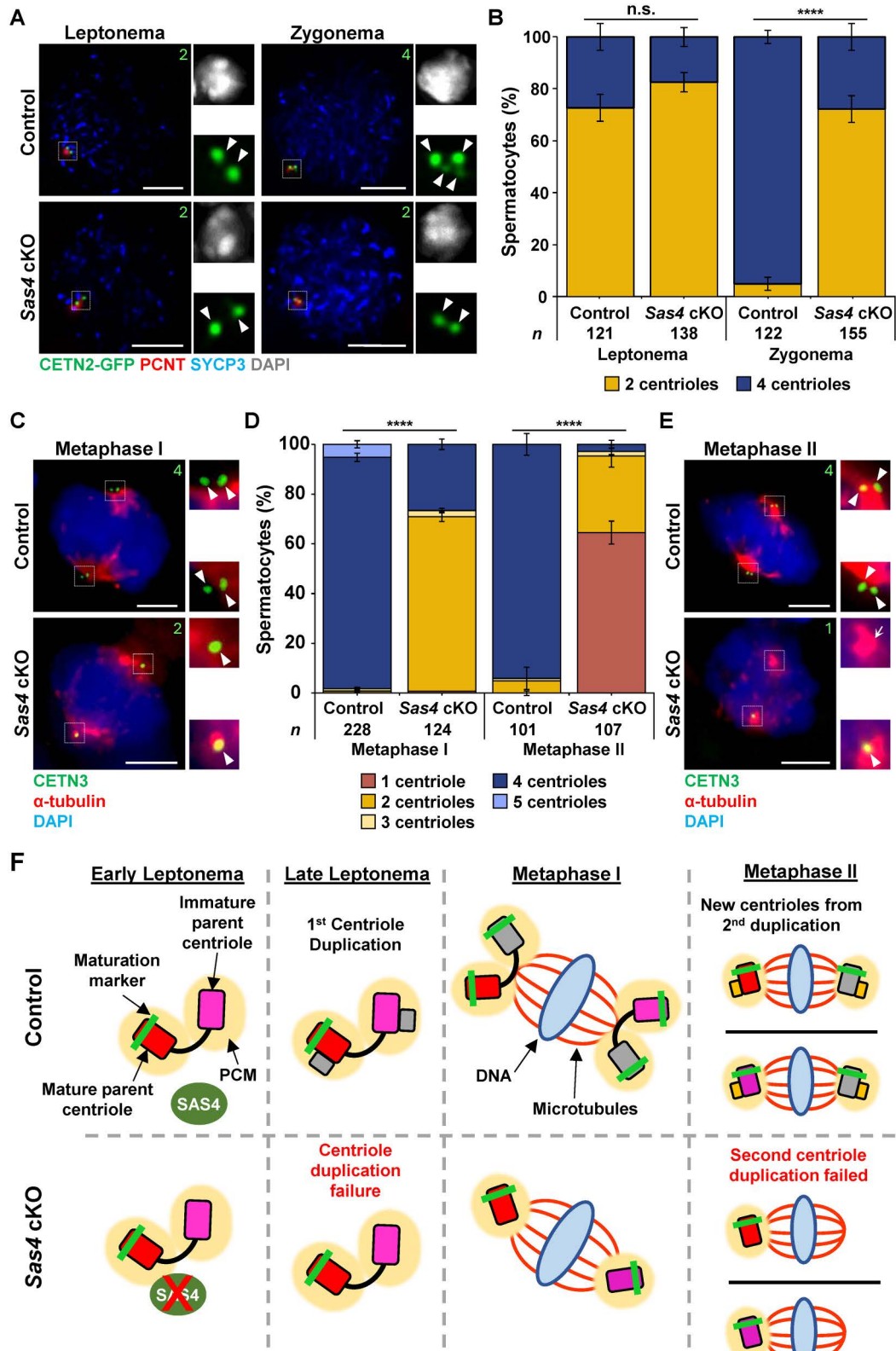

**Fig 3. Spermatocytes successfully form bipolar spindles and complete chromosome segregation despite centriole duplication failure. A.** Leptonema and zygonema stage control and *Sas4* cKO spermatocytes from 13 dpp mice expressing CETN2-GFP were immunolabeled against PCNT (red)

and SYCP3 (blue) and stained with DAPI (grey outset). Zoomed images of the centrioles are outset to the right of the corresponding images. The white arrowheads indicate the centrioles. The green number indicates the total number of centrioles per cell. Scale bars = 5 μm. **B.** Quantification of centriole foci observed during leptonema and zygonema in both control and *Sas4* cKO spermatocytes. The total number of cells quantified for control and *Sas4* cKO mice were 243 and 293, respectively. P values for leptonema and zygonema were 0.0771 and < 0.0001, respectively. Error bars show mean ± SEM. P values obtained from two-tailed Student's t-test. n.s. (not significant), ****P < 0.0001. **C.** Metaphase I control and *Sas4* cKO spermatocytes from 23-27 dpp mice were immunolabeled against CETN3 (green), α-tubulin (red) and stained with DAPI (blue). Zoomed images of the centrioles are outset to the right of the corresponding images. The white arrowheads indicate the centrioles. The green number indicates the total centrioles per cell. Scale bars = 5 μm. **D.** Quantification of centriole foci observed during metaphase I and metaphase II in both control and *Sas4* cKO spermatocytes. Immunolabeling was performed on ≥ 3 biological replicates, with ≥ 30 spermatocytes quantified per replicate. The total number of cells quantified for control and *Sas4* cKO mice were 329 and 231, respectively. P values for metaphase I and metaphase II were < 0.0001 and < 0.0001, respectively. Error bars show mean ± SEM. P values obtained from two-tailed Student's t-test. ****P < 0.0001. **E.** Metaphase II control and *Sas4* cKO spermatocytes from 23-27 dpp mice were immunolabeled against CETN3 (green), α-tubulin (red) and stained with DAPI (blue). Zoomed images of the centrioles are outset to the right of the corresponding images. The white arrowheads indicate the centrioles. The white arrow indicates the acentriolar spindle pole. The green number indicates the total centrioles per cell. Scale bars = 5 μm. **F.** Diagram illustrating how *Sas4* cKO leads to centriole duplication failure during meiosis. As a result, metaphase I spermatocytes were observed to harbor one centriole at each bipolar spindle pole and metaphase II spermatocytes were observed to harbor a single centriole at one of the two bipolar spindle poles. Red rectangle with rounded corners = mature parent centriole, pink rectangle with rounded corners = immature parent centriole, green bar = maturation marker, yellow oval = PCM, green oval = SAS4, red X = indicates depletion, grey rectangle with rounded corners = new centriole from first centriole duplication, yellow rectangle with rounded corners = new centrioles from second centriole duplication, blue oval = DNA, red lines = microtubules.

excision of the *Sas4* floxed allele or there was residual SAS4 protein that was sufficient to enable centriole duplication during meiotic prophase.

The *Sas4* cKO spermatocytes that did contain 4 centrioles showed no difference in morphology when compared to control spermatocytes. All *Sas4* cKO metaphase I spermatocytes, regardless of whether they harbored 2 or 4 centrioles, were capable of undergoing separation to support the formation of a bipolar spindle. Therefore, 70% of *Sas4* cKO spermatocytes had a single centriole at each pole during chromosome segregation in meiosis I (Fig 3C, 3D, and 3F). However, by metaphase II, 65% of secondary spermatocytes formed a bipolar spindle and harbored only one centriole, and 31% harbored only 2 centrioles, one at each spindle pole (Fig 3D-3F). These results also indicate that the second centriole duplication failed in 96% of *Sas4* cKO spermatocytes, suggesting the excision of the *Sas4* floxed allele was efficient in the majority of *Sas4* cKO spermatocytes. Attempts to demonstrate protein depletion in *Sas4* cKO spermatocytes via IF were unsuccessful. Experiments performed using two separate antibodies against SAS4 (Proteintech 11517-1-AP and Affinity DF2313) resulted in broad cytoplasmic staining that was not specific to the protein of interest in both control and *Sas4* cKO spermatocytes. Despite our efforts, the specific phenotypes we observed relating to centriole duplication failure and male infertility suggest the mouse model successfully led to SAS4 depletion. The *Sas4* cKO mouse line is a commercially available model that others have previously characterized and demonstrated to be a robust system for SAS4 depletion [46,62,63]. Furthermore, the *Spo11*-Cre mouse line we utilized to generate the *Sas4* cKO during early meiosis has been well established and used in house with other cKO alleles with efficient Cre excision [26,48,50,51].

Further characterization of the 33% of *Sas4* cKO metaphase II spindle poles that were acentriolar was performed by measuring both the length and width of the spindle MTs (see "Materials and methods" and S1 Data). The acentriolar spindle pole in *Sas4* cKO spermatocytes was observed to be reduced in both length (acentriolar pole to centriolar pole ratio = 0.643) and width (acentriolar pole to centriolar pole ratio = 0.802). In contrast, metaphase II spermatocytes with two centriolar spindle poles did not have differences in length or width compared to one another (length ratio = 0.920; width ratio = 0.944). Despite centriole duplication failure, histological assessment of *Sas4* cKO testes demonstrated that round spermatid formation was equivalent to littermate controls (Fig 1D), which indicated that meiosis II was successfully completed.

## Centriole maturation is unaffected by SAS4 depletion

In addition to centriole duplication, another important aspect of centriole biogenesis is centriole maturation. During this process, distal and subdistal appendages are loaded to stabilize the new centriole [62,63]. Without this stabilization, the

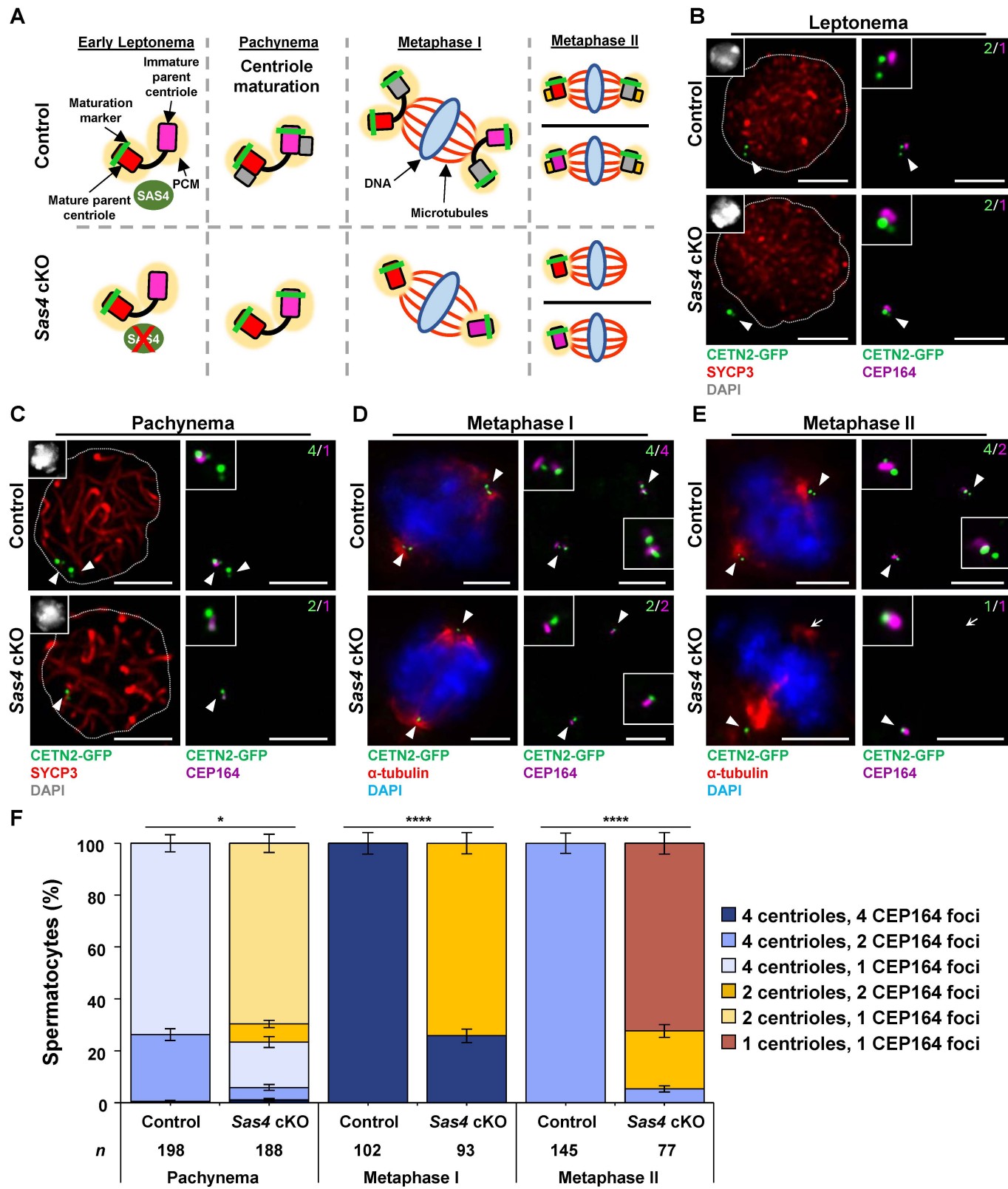

**Fig 4. Centriole maturation is unaffected by SAS4 depletion. A.** Diagram illustrating how *Sas4* cKO does not affect centriole maturation. As a result, all observed centrioles in *Sas4* cKO spermatocytes are mature by metaphase I and metaphase II. Red rectangle with rounded corners = mature parent

centriole, pink rectangle with rounded corners = immature parent centriole, green bar = maturation marker, yellow oval = PCM, green oval = SAS4, red X = indicates depletion, grey rectangle with rounded corners = new centriole from first centriole duplication, yellow rectangle with rounded corners = new centrioles from second centriole duplication, blue oval = DNA, red lines = microtubules. **B.** Leptotene stage control and *Sas4* cKO spermatocytes from 12 dpp mice harboring CETN2-GFP (green) were immunolabeled against SYCP3 (red) and CEP164 (purple) and stained with DAPI (grey inset). Zoomed images of the centrioles are inset on corresponding images. The white arrowheads indicate the centrosome. The dashed line outlines the DAPI signal. The green number indicates the total centrioles per cell. The magenta number indicates the total number of CEP164 foci per cell. Scale bars = 5 μm. **C.** Pachytene stage control and *Sas4* cKO spermatocytes from 23-27 dpp mice harboring CETN2-GFP (green) were immunolabeled against SYCP3 (red) and CEP164 (purple) and stained with DAPI (grey inset). Zoomed images of the centrioles are inset on corresponding images. The white arrowheads indicate the centrosome. The dashed line outlines the DAPI signal. The green number indicates the total centrioles per cell. The magenta number indicates the total number of CEP164 foci per cell. Scale bars = 5 μm. **D.** Metaphase I control and *Sas4* cKO spermatocytes from 23-27 dpp mice harboring CETN2-GFP (green) were immunolabeled against α-tubulin (red) and CEP164 (purple) and stained with DAPI (blue). Zoomed images of the centrioles are inset on corresponding images. The white arrowheads indicate the centrosome. The green number indicates the total centrioles per cell. The magenta number indicates the total number of CEP164 foci per cell. Scale bars = 5 μm. **E.** Metaphase II control and *Sas4* cKO spermatocytes from 23-27 dpp mice harboring CETN2-GFP (green) were immunolabeled against α-tubulin (red) and CEP164 (purple) and stained with DAPI (blue). Zoomed images of the centrioles are inset on corresponding images. The white arrowheads indicate the centrosome. The white arrows indicate non-centrosomal MTOC (ncMTOC). The green number indicates the total centrioles per cell. The magenta number indicates the total number of CEP164 foci per cell. Scale bars = 5 μm. **F.** Quantification and localization of CEP164 foci in relation to centrioles in control and *Sas4* cKO pachytene, metaphase I, and metaphase II stage spermatocytes. Immunolabeling was performed on 3 biological replicates with ≥ 30 spermatocytes quantified per replicate. The total number of cells quantified for control and *Sas4* cKO mice were 447 and 357, respectively. P values for pachynema, metaphase I, and metaphase II were 0.0179, < 0.0001, and < 0.0001, respectively. Error bars show mean ± SEM. P values obtained from two-tailed Student's t-test. *$P < 0.05$, ****$P < 0.0001$.

centriole remains immature and prone to disassembly [51,62,63]. We assessed spermatocytes for maturation by immunolabeling against the subdistal appendage CEP164. A primary spermatocyte enters meiosis with a centrosome that contains one mature parent centriole and one immature parent centriole (Fig 4A and 4B). Once duplication occurs during late leptonema, there is still only the single mature parent centriole, one immature parent centriole, and two immature daughter centrioles (Fig 4A and 4B). During pachynema, centriole maturation processes begin and the original immature parent centriole undergoes maturation, resulting in two mature parent and two immature daughter centrioles (Fig. 4A and 4C). Then, by metaphase I, the remaining daughter centrioles also complete maturation, resulting in metaphase I spermatocytes with four mature centrioles (Fig 4A and 4D). Therefore, after meiosis I is completed, each secondary spermatocyte inherits two mature centrioles (Fig 4A). During interkinesis, secondary spermatocytes undergo the second round of centriole duplication (Fig 4A). However, the newly formed centrioles remain immature, never acquiring the addition of distal and subdistal appendages. During meiosis II, the two centrosomes, both consisting of one mature and one immature centriole, separate to form bipolar spindles (Fig 4A and 4E; [26]). Therefore, each round spermatid inherits one mature and one immature centriole that are both critical for the HTCA and flagella formation during spermiogenesis (Fig 4A; [4,26,29]). Although *Sas4* cKO spermatocytes failed to duplicate centrioles, the immature centriole that is present at meiotic entry successfully underwent maturation during meiotic prophase (Fig 4A-F). Therefore, SAS4 depletion does not inhibit centriole maturation during spermatogenesis. At all subsequent stages, centrioles were always classified as mature, which suggests that centriole duplication failure driven by SAS4 depletion does not cause destabilization of distal or subdistal appendages (Fig 4A-F).

## Secondary spermatocytes utilize a non-centrosomal MTOC for bipolar spindle formation

Having observed bipolar *Sas4* cKO secondary spermatocytes with a single centriole, we aimed to identify what factors were coordinating MT assembly at the spindle pole without a centriole. In oocytes, because centrioles are disassembled, chromosome segregation is mediated by aMTOCs [53]. These aMTOCs are comprised of many proteins that are also found to be a part of the PCM in the centrosomes of spermatocytes [53]. We sought to determine if the same PCM proteins were present at this acentriolar pole in *Sas4* cKO metaphase II spermatocytes. We assessed the PCM components CEP192 and GCP2. CEP192 is an important scaffold protein that aids in centriole stabilization and recruitment of PLK4 to the centriole at the time of duplication [64–67]. GCP2 is a component of the γ-tubulin ring complex (γ-TURC) that serves as the canonical site for microtubule nucleation at the spindle pole [68]. Both CEP192 and GCP2 are observed to localize

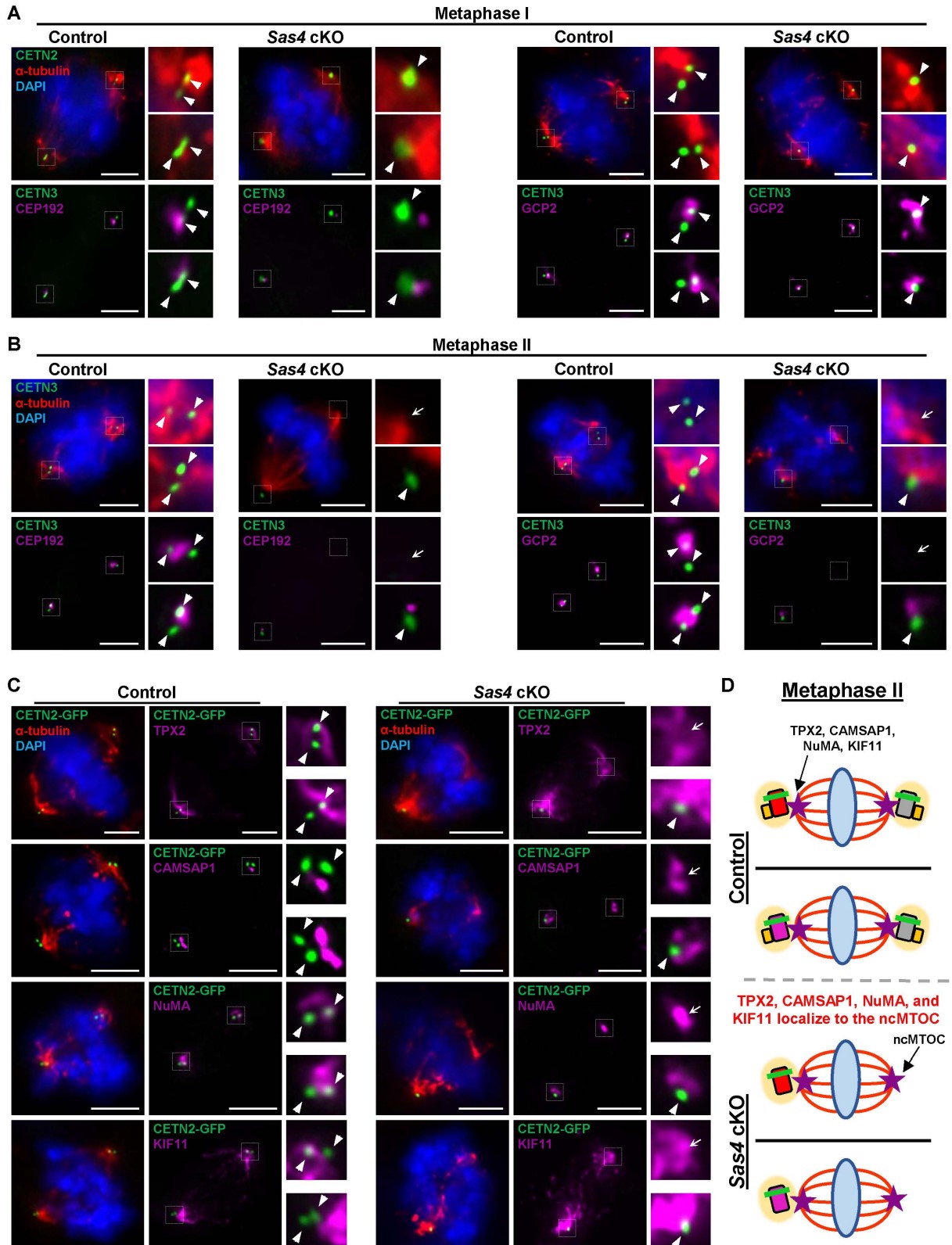

**Fig 5. Secondary spermatocytes utilize a ncMTOC for bipolar spindle formation. A.** Metaphase I control and *Sas4* cKO spermatocytes from 23-27 dpp mice were immunolabeled against CETN3 (green), α-tubulin (red), and CEP192, or GCP2 (purple) and stained with DAPI (blue). Zoomed images

of the centrioles are outset to the right of the corresponding images. The white arrowheads indicate the centrosome. Scale bars = 5 µm. **B.** Metaphase II control and *Sas4* cKO spermatocytes from 23-27 dpp mice were immunolabeled against CETN3 (green), α-tubulin (red), and CEP192, or GCP2 (purple) and stained with DAPI (blue). Zoomed images of the centrioles are outset to the right of the corresponding images. The white arrowheads indicate the centrosome. The white arrows indicate the ncMTOC. Scale bars = 5 µm. **C.** Metaphase II control and *Sas4* cKO spermatocytes from 23-27 dpp mice expressing CETN2-GFP were immunolabeled against α-tubulin (red), TPX2, CAMSAP1, NuMA, and KIF11 (purple) and stained with DAPI (blue). Zoomed images of the centrioles are outset to the right of the corresponding images. The white arrowheads indicate the centrosome. The white arrows indicate the ncMTOC. Scale bars = 5 µm. **D.** Diagram illustrating the localization of MT associated factors to the centrosome and ncMTOC during metaphase II in control and *Sas4* cKO spermatocytes. Red rectangle with rounded corners = original mature parent centriole, pink rectangle with rounded corners = original immature parent centriole, grey rectangle with rounded corners = new centriole from first centriole duplication, yellow rectangle with rounded corners = new centrioles from second centriole duplication, green bar = maturation marker, yellow oval = PCM, purple star = MT associated factors, blue oval = DNA, red lines = microtubules.

to the centrosome in control primary and secondary spermatocyte spindle poles (Fig 5A and 5B). While both proteins localized to *Sas4* cKO spermatocyte spindle poles with 1 or 2 centrioles, neither CEP192 nor GCP2 was observed at the acentriolar pole in secondary spermatocytes (Fig 5A and 5B). This lack of centrosomal proteins at the acentriolar spindle poles in *Sas4* cKO secondary spermatocytes suggests they are non-centrosomal MTOCs (ncMTOCs).

Next, we wanted to identify the proteins responsible for coordinating the formation of the observed ncMTOCs in *Sas4* cKO secondary spermatocytes. To coordinate spindle assembly, MTs must successfully undergo nucleation and extension, avoid disassembly (catastrophe), and be properly focused and oriented into two separate bipolar spindles. We assessed four MT-associated proteins that have known roles in coordinating these MT dynamics. The first was Calmodulin Regulated Spectrin Associated Protein 1 (CAMSAP1), a MT stabilizing protein that binds the minus ends of MTs to deter catastrophe [41,69]. We also assessed the microtubule nucleation and stabilizing factor, Targeting Protein for Xklp2 (TPX2), which aids in preventing MT catastrophe by binding more evenly along the entire lengths of the MTs [70,71]. Both CAMSAP1 and TPX2 have also been observed to promote MT nucleation in the absence of γ-TURCs [69,70,72–74]. Additionally, we assessed Nuclear Mitotic Apparatus Protein 1 (NuMA) localization, which has known roles in focusing MT minus ends to a uniform location while also facilitating the proper bipolar spindle pole orientation with respect to the cellular membrane [75,76]. Finally, we examined Kinesin Family Member 11 (KIF11, also known as Eg5), which is a MT motor protein that drives spindles poles apart during MTOC separation [77–79].

CAMSAP1, TPX2, NuMA, and KIF11 all localize to the centrosomal spindle poles in both the control and *Sas4* cKO secondary spermatocytes (Fig 5C and 5D). In addition, all four proteins localized to the ncMTOC in *Sas4* cKO secondary spermatocytes (Fig 5C and 5D). These results corroborate with a previous observation made when assessing *Plk4* cKO spermatocytes. This study showed that CAMSAP1–3, TPX2, NuMA, and KIF11 localized to the centriolar poles and ncMTOC that arose in *Plk4* cKO spermatocytes during meiotic divisions [51]. We also observed no difference in the prevalence of chromosome missegregation events between *Sas4* cKO and control spermatocytes during anaphase I and anaphase II, suggesting that spermatocytes with a ncMTOC successfully complete chromosome segregation (Fig 6A and 6B). Taken together, this data indicates that these MT-associated factors present at the ncMTOC are key factors required for mediating chromosome segregation during meiosis in mammalian spermatocytes and demonstrates that canonical centrosomal components are not essential for this process.

## Aberrant spermiogenesis is observed in spermatids that inherit less than two centrioles

Haploid round spermatids canonically inherit two centrioles after completing the second meiotic division (Fig 6C-E). However, *Sas4* cKO round spermatids are observed to inherit either 1 or 0 centrioles (Fig 6C-E). Despite this reduced centriole inheritance, round spermatids formed at normal rates in *Sas4* cKO mice and continue to develop to an early elongated stage (Figs 1D and 6C-E). However, spermiogenesis processes in the *Sas4* cKO spermatids are defective.

During spermiogenesis, spermatids undergo the transition from round to elongated spermatids [28]. To assess this process in control and *Sas4* cKO spermatids, we evaluated several key features, including DNA morphology, acrosome

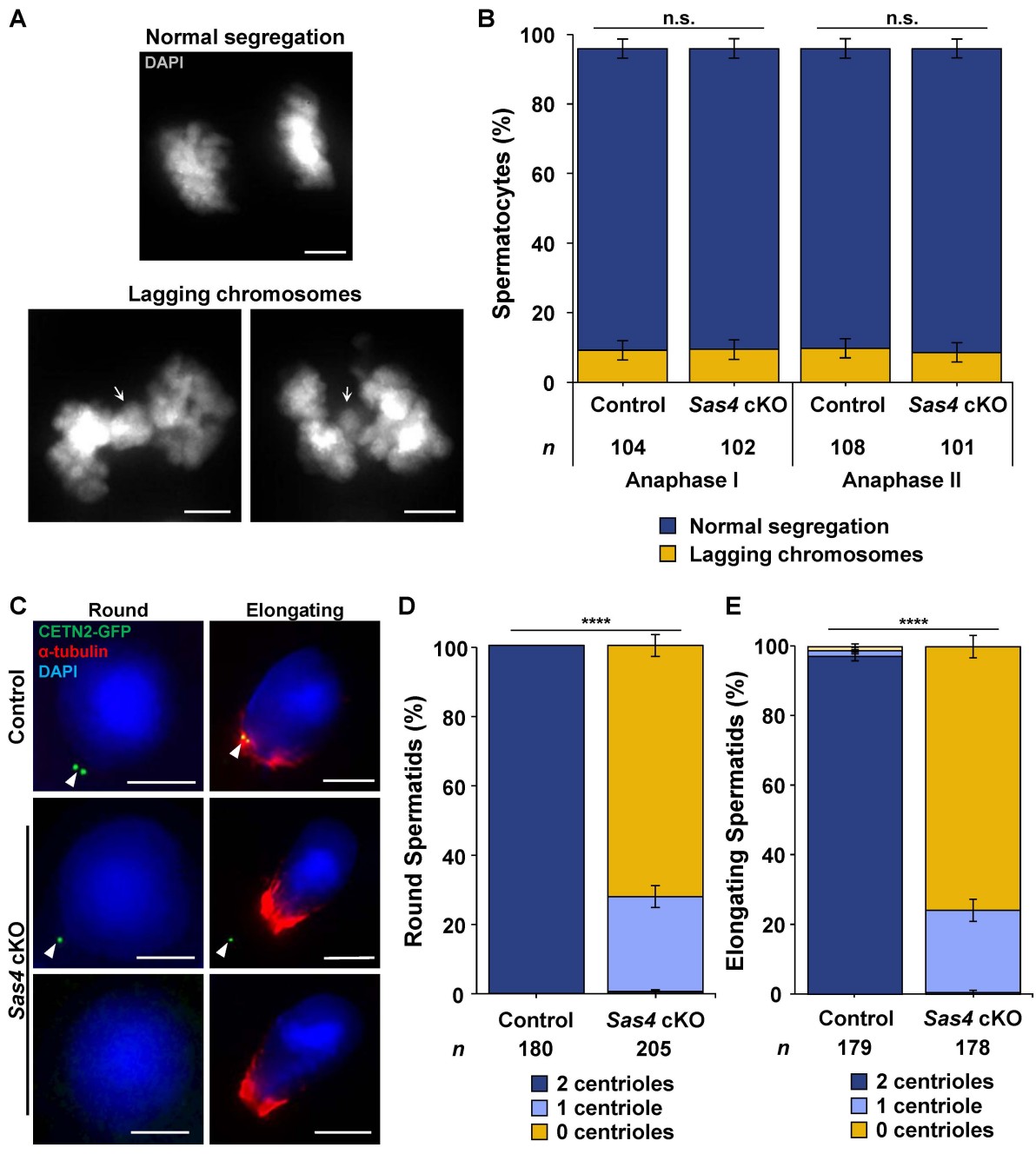

**Fig 6. Round and elongating spermatids form with less than two centrioles. A.** Representative images of normal chromosome segregation and chromosome missegregation events during anaphase II spermatocytes stained with DAPI. The white arrows indicate the lagging chromosome event. Scale bars = 5 μm. **B.** Quantification of lagging chromosomes in anaphase I and II spermatocytes. Immunolabeling was performed on 5 biological replicates with ≥ 20 spermatocytes quantified per replicate. The total number of cells quantified for control and *Sas4* cKO mice was 212 and 203, respectively. P values for anaphase I and anaphase II were 0.9655 and 0.7566, respectively. Error bars show mean ± SEM. P values obtained from two-tailed Student's t-test. n.s. (not significant). **C.** Control and *Sas4* cKO round and elongating spermatids from 28-36 dpp mice expressing CETN2-GFP (green) were immunolabeled against α-tubulin (red) and stained with DAPI (blue). The white arrowheads indicate the centrioles. Scale bars = 5 μm. **D.** Quantification of CETN2-GFP foci in control and *Sas4* cKO round spermatids. Immunolabeling was performed on 3 biological replicates with ≥ 50 spermatocytes quantified per replicate. The total number of cells quantified for control and *Sas4* cKO mice were 180 and 205, respectively. P value = < 0.0001. Error bars show mean ± SEM. P values obtained from two-tailed Student's t-test. ****P < 0.0001. **E.** Quantification of CETN2-GFP foci in control and *Sas4* cKO

elongating spermatids. Immunolabeling was performed on 3 biological replicates with ≥50 spermatocytes quantified per replicate. The total number of cells quantified for control and *Sas4* cKO mice were 179 and 178, respectively. P value =<0.0001. Error bars show mean±SEM. P values obtained from two-tailed Student's t-test. ****P<0.0001.

development, perinuclear ring formation, and manchette assembly and disassembly [28,41]. We utilized DAPI stain to visualize DNA morphology, and immunolabeling against lectin-PNA to assess the formation of the acrosome [80]. Additionally, we immunolabeled against the centriolar satellite protein Coiled-coil Domain-containing Protein 13 (CCDC13), which has been previously reported to localize to the perinuclear ring and the centrosome in elongating spermatids [51,81]. Immunolabeling against α-tubulin was conducted to visualize the manchette structure [39]. We also assessed later-stage elongating spermatids for flagella formation by immunolabeling against δ-tubulin and acetylated tubulin [42,51].

Spermiogenesis is characterized by 16 steps that can be broken down into four major phases: Golgi phase, cap phase, tail phase, and maturation phase [82]. During the Golgi phase (steps 1–3), control spermatids begin to develop polarity (Fig 7A; [82]. The Golgi apparatus, situated on the side of the spermatid that will become the proximal end, begins producing enzymes such as glycohydrolases, proteases, and phosphatases that will make up the acrosome contents [28,41,83]. These components are important for the sperm-egg interaction that triggers the release of the acrosome contents (acrosome reaction) in order for fertilization to occur [82,83]. During the cap phase (steps 4–7), the acrosome spreads to cover approximately half of the nucleus (Fig 7A and 7B; [82,83]. During this period, DNA transcription is inactivated and the chromatin becomes highly condensed [82]. In the tail phase (steps 8–12), flagella formation takes place [82]. DNA morphology also begins transitioning from round (steps 1–8) to elongated starting in step 9 (Fig 7A). These processes are facilitated by the transient protein trafficking-related structures known as the manchette and perinuclear ring [41,84]. Manchette MTs polymerize from the centriole adjunct at the distal end of the spermatid head extending towards the perinuclear ring starting at step 9 (Fig 7C; [40,41]). As DNA remodeling progresses to form a point (step 10) that continues to become a pronounced hook structure characteristic of mouse spermatocytes (steps 11–12), the manchette MTs begin to shorten (steps 11–12; Fig 7C and 7D; [28,41]). CCDC13 localizes to the perinuclear ring structure around the hemisphere of the spermatid beginning at step 9 and is maintained during DNA remodeling (steps 10–12; Fig 7E and 7F; [28,41]). CCDC13 also localizes as a single focus in the neck region of the spermatids beginning at step 9 and remaining for the rest of spermiogenesis (Fig 7E). In the final steps of spermiogenesis or maturation phase (steps 13–16), the manchette and perinuclear ring are disassembled, DNA condensation to a compact hook is completed, the acrosome fully expands to cover the proximal hemisphere of the spermatid, and excess cytoplasm is removed via phagocytosis into the surrounding Sertoli cells in the testis [28,41,82].

*Sas4* cKO spermatids exhibited normal morphology and spermiogenesis progression from steps 1–8 (Fig 7A, 7C, and 7E). However, while DNA morphology at step 9 remained unchanged, the formation of the acrosome was perturbed (Fig 7A-F). This is exemplified by the ~35% reduction in acrosome width at step 9 in the *Sas4* cKO spermatids compared to the control (Fig 7A and 7B). By the maturation phase of spermiogenesis, *Sas4* cKO spermatids have misshapen acrosome and DNA morphology (Fig 7A). Defects in the manchette and perinuclear ring arise as well. The length of the *Sas4* cKO spermatid manchette MTs was ~60% longer in steps 11–12 than control spermatid manchette MTs (Fig 7C and 7D). Additionally, manchette MTs were never dismantled in *Sas4* cKO spermatids at steps 13–16 (Fig 7C). The perinuclear ring, which helps stabilize the plus ends of the manchette MTs displayed a decrease in diameter. In step 10 *Sas4* cKO spermatids, the diameter of the perinuclear ring was 5.81 µm compared to 9.35 µm in control step 10 spermatids (Fig 7E and 7F). This observation aligns with the decrease in average manchette width in step 11–12 *Sas4* cKO spermatids compared to the control (5.18 µm and 9.27 µm, respectively).

We also assessed CCDC13 localization at the neck in spermatids. When SAS4 was depleted, we only observed 32% of step 9 spermatids and 17% of step 10 spermatids with CCDC13 localized to the neck region (Fig 7E and 7G). By the

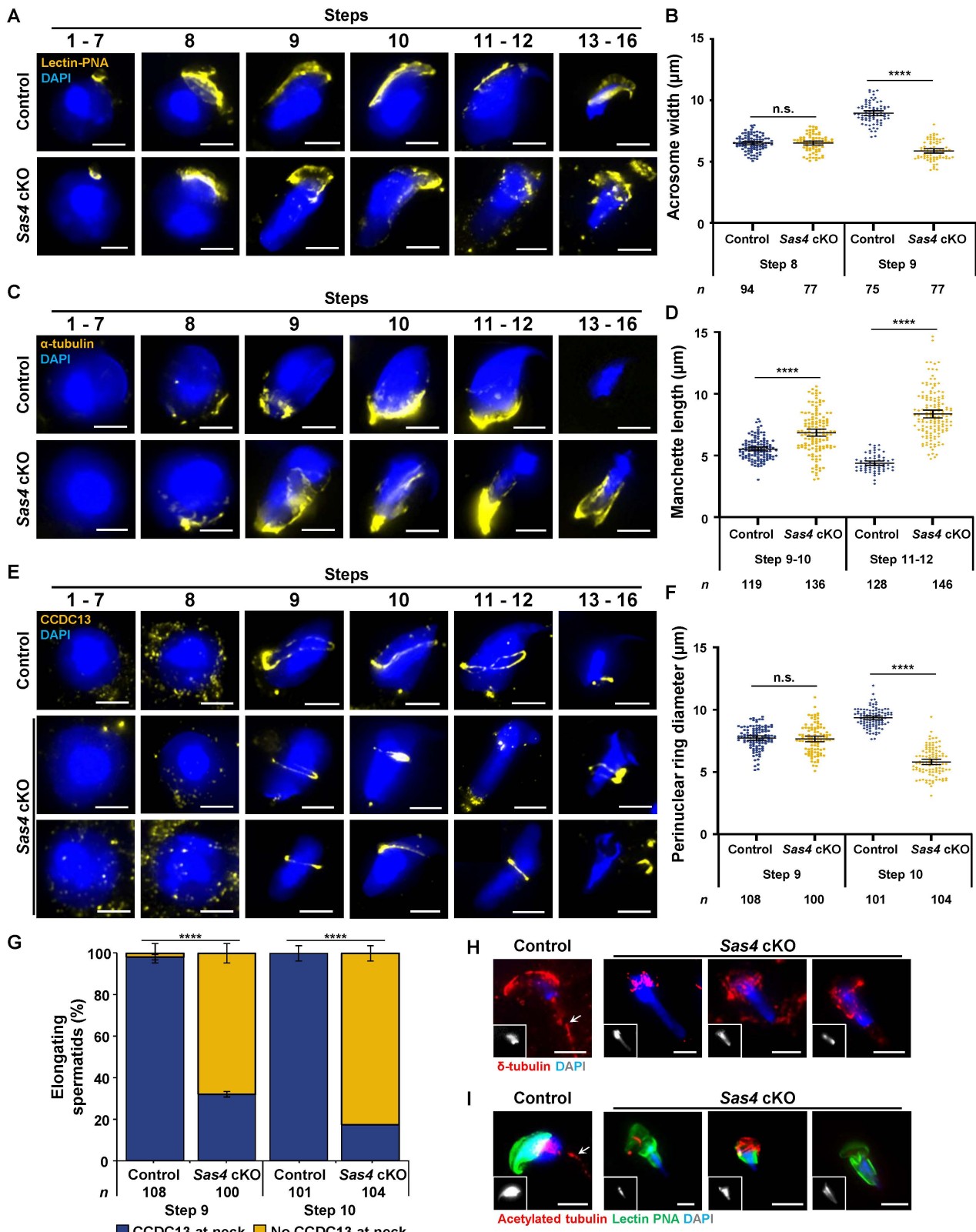

**Fig 7. Aberrant spermiogenesis is observed in spermatids that inherit less than two centrioles. A.** Representative images of control and *Sas4* cKO spermatids at progressive steps of spermiogenesis from 27-36 dpp mice. Spermatids were immunolabeled against lectin-PNA (yellow) and stained

with DAPI (blue). Immunolabeling against lectin-PNA allowed for the characterization of the acrosome during spermiogenesis, which is a structure that develops to cover the proximal end of the spermatid head and is necessary for fertility. Scale bars = 5 µm. **B.** Quantification of acrosome diameter in control and *Sas4* cKO spermatids at steps 8 and 9 of spermiogenesis. Immunolabeling against lectin-PNA and measurement of acrosome diameter was performed on 3 biological replicates with ≥ 15 spermatocytes quantified per replicate. The average diameter of the acrosome in step 8 spermatids was 6.51 µm and 6.50 µm in control and *Sas4* cKO mice, respectively. The average diameter of the acrosome in step 9 spermatids was 8.95 µm and 5.87 µm in control and *Sas4* cKO mice, respectively. The total number of cells quantified for control and *Sas4* cKO mice were 169 and 154, respectively. P values for step 8 and step 9 acrosome diameters were 0.8947 and < 0.0001, respectively. Error bars show mean ± 95% CI. P values obtained from two-tailed Student's t-test. n.s. (not significant), ****P < 0.0001. **C.** Representative images of control and *Sas4* cKO spermatids at progressive steps of spermiogenesis from 27-36 dpp mice. Spermatids were immunolabeled against α-tubulin (yellow) and stained with DAPI (blue). Immunolabeling against α-tubulin allowed for assessment of the manchette morphology, a MT based protein trafficking structure critical for successful cellular remodeling during spermiogenesis. Scale bars = 5 µm. **D.** Quantification of manchette MT length in control and *Sas4* cKO spermatids at steps 9-10 and 11-12 of spermiogenesis. Immunolabeling against α-tubulin and measurement of MT length was performed on 3 biological replicates with ≥ 30 spermatocytes quantified per replicate. The average length of manchette MTs in step 9-10 spermatids was 5.53 µm and 6.84 µm in control and *Sas4* cKO mice, respectively. The average length of manchette MTs in step 11-12 spermatids was 3.94 µm and 8.37 µm in control and *Sas4* cKO mice, respectively. The total number of cells quantified for control and *Sas4* cKO mice were 247 and 284, respectively. P values for step 9-10 and step 11-12 MT lengths were <0.0001 and <0.0001, respectively. Error bars show mean ± 95% CI. P values obtained from two-tailed Student's t-test. ****P < 0.0001. **E.** Representative images of control and *Sas4* cKO spermatids at progressive steps of spermiogenesis from 27-36 dpp mice. Spermatids were immunolabeled against CCDC13 (yellow) and stained with DAPI (blue). CCDC13 is a protein shown to localize to the perinuclear ring in elongating spermatids. The perinuclear ring helps stabilize the manchette MTs and promote successful protein trafficking. CCDC13 is also associated with the centrosome and localizes to the neck region in spermatids. Assessment of the perinuclear ring and CCDC13 localization to the neck region helped determine the fidelity of the protein trafficking in spermatids and indicate the attachment of the tail to the spermatid head, respectively. Scale bars = 5 µm. **F.** Quantification of perinuclear ring diameter in control and *Sas4* cKO spermatids at steps 9 and 10 of spermiogenesis. Immunolabeling against CCDC13 and measurement of perinuclear ring diameter was performed on 3 biological replicates with ≥ 30 spermatocytes quantified per replicate. The average diameter of the perinuclear ring in step 9 spermatids was 7.75 µm and 7.66 µm in control and *Sas4* cKO mice, respectively. The average diameter of the perinuclear ring in step 10 spermatids was 9.35 µm and 5.81 µm in control and *Sas4* cKO mice, respectively. The total number of cells quantified for control and *Sas4* cKO mice were 208 and 205, respectively. P values for step 9 and step 10 acrosome diameters were 0.4548 and < 0.0001, respectively. Error bars show mean ± 95% CI. P values obtained from two-tailed Student's t-test. n.s. (not significant), ****P < 0.0001. **G.** Quantification of CCDC13 localization to the neck region in control and *Sas4* cKO spermatids at steps 9 and 10 of spermiogenesis. Immunolabeling against CCDC13 was performed on 3 biological replicates with ≥ 30 spermatocytes quantified per replicate. The percentage of spermatids with CCDC13 foci in the neck region in step 9 was 98% and 32% in control and *Sas4* cKO mice, respectively. The percentage of spermatids with CCDC13 foci in the neck region in step 10 was 100% and 17% in control and *Sas4* cKO mice, respectively. The total number of cells quantified for control and *Sas4* cKO mice were 209 and 204, respectively. P values for step 9 and step 10 spermatids were ****P < 0.0001 and ****P < 0.0001, respectively. Error bars show mean ± SEM. P values obtained from two-tailed Student's t-test. ****P < 0.0001. **H.** Control and *Sas4* cKO spermatozoa were immunolabeled against δ-tubulin (red) and stained with DAPI (blue, and a grey inset in the lower left corner to highlight DNA morphology). The white arrow indicates the flagella. Scale bars = 5 µm. **I.** Control and *Sas4* cKO spermatozoa were immunolabeled against lectin-PNA (green), acetylated tubulin (red), and stained with DAPI (blue, and a grey inset in the lower left corner to highlight DNA morphology). The white arrow indicates the flagella. Scale bars = 5 µm.

maturation phase, all *Sas4* cKO spermatids had lost CCDC13 localization to the neck (Fig 7E). We hypothesize that the *Sas4* cKO spermatids observed to have CCDC13 present in the neck region inherited a single mature centriole (Fig 7E). In these spermatids, even though MT extension from the centriole to form the flagella was possible, the immature centriole that facilitates the head-tail attachment was absent. By steps 13–16, when the CCDC13 signal is no longer visible in any *Sas4* cKO spermatocytes, it is likely that the tail was no longer capable of maintaining association with the head (Fig 7E). This inability to maintain the head-tail attachment is likely also why we observed approximately 75% of round spermatids with 0 centrioles (Fig 6A and 6B). Without the immature centriole, the centrosome attachment to the nucleus is subject to failure as early as steps 1–4 of spermiogenesis. In *Sas4* cKO spermatids that do not inherit any centrioles, flagella formation was never able to initiate. Assessment of the presence of flagella in maturation phase spermatids was performed utilizing immunolabeling of δ-tubulin and acetylated tubulin [42,51,85]. An attached tail can clearly be visualized by δ-tubulin and acetylated tubulin localization in proximity with DAPI staining in control spermatids (Fig 7H and 7I). However, flagella were not observed in *Sas4* cKO spermatids (Fig 7H and 7I). Co-staining with DAPI and immunolabeling with lectin-PNA also demonstrate *Sas4* cKO spermatids had perturbed DNA and acrosome morphology, as previously observed (Fig 7H and 7I).

In summary, the defects in *Sas4* cKO spermatids undergoing spermiogenesis are present initially at step 9, when the aberrant acrosome, perinuclear ring, and manchette formation are evident (Fig 7A, 7C, and 7E). Centrioles play

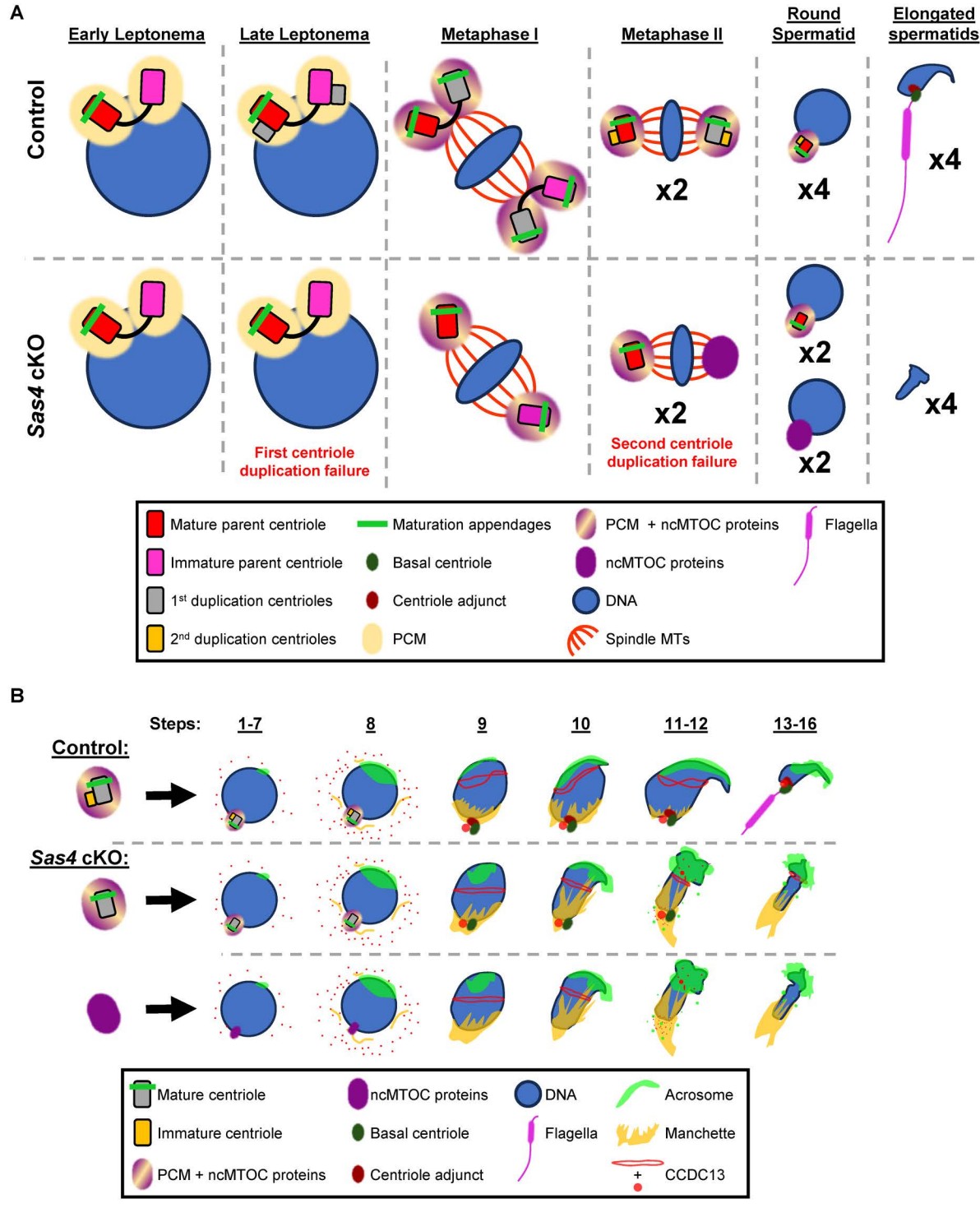

**Fig 8. Spermatids require two centrioles for successful spermiogenesis. A.** Diagram illustrating centriole duplication in control and *Sas4* cKO spermatocytes during spermatogenesis. In control spermatocytes, centrioles undergo duplication during late leptonema. By metaphase I, each spindle pole contains two mature centrioles and both PCM MT-associated proteins. By metaphase II, each spindle pole now only contains one mature centriole and one immature centriole in addition to the PCM and MT-associated proteins. Each resulting round spermatid inherits a complete centrosome with one mature and one immature centriole that will progress after meiosis to form fully mature spermatids. This is presented as a remodeled head structure and attached flagella. In the *Sas4* cKO spermatocytes, the first centriole duplication fails, leading to metaphase I spermatocytes with only one

mature centriole at each bipolar spindle pole (PCM and ncMTOC proteins are also present). The second centriole duplication event also fails, leading to metaphase II spermatocytes with one mature centriole at one spindle pole and no centriole at the other pole. At the spindle pole without a centriole, there are only ncMTOC proteins, not PCM proteins. This leads to half of *Sas4* cKO round spermatids inheriting a single mature centriole and half inheriting no centrioles. As a result, spermatid remodeling is unsuccessful and leads to mishappen head formation and a lack of flagella. Red rectangle with rounded corners = original mature parent centriole, pink rectangle with rounded corners = original immature parent centriole, grey rectangle with rounded corners = new centrioles from first centriole duplication event, yellow rectangle with rounded corners = new centrioles from second centriole duplication event, green bar = maturation marker, green oval = basal centriole, dark red oval = centriole adjunct, yellow oval = PCM, yellow and purple oval = PCM and MT associated proteins, purple oval = microtubule-associated proteins, blue = DNA, red lines = microtubules, orange = flagella. **B.** Diagram illustrating the progression of control and *Sas4* cKO spermatids through spermiogenesis. Control round spermatids inherit two centrioles, one mature and one immature. As they progress through spermiogenesis, they undergo nuclear remodeling and DNA compaction. This process is mediated by the transient protein trafficking structure present in step 9-12 spermatids, the manchette. The manchette MTs are stabilized at their plus ends by the perinuclear ring that forms at the hemisphere of the spermatid head and is visible during steps 9 to 12. The centriole adjunct, formed from the immature centriole, serves as both the nucleation site for the manchette MTs and facilitates the head-tail attachment. The mature centriole serves as the base for flagella axoneme extension. Throughout this process, the spermatid also develops an acrosome consisting of proteins necessary for fertilization. The acrosome proteins first accumulate at a localized region at the proximal end of the spermatid (steps 1-7) that continues to expand to cover the entire top half of the sperm head by the end of spermiogenesis (steps 8-16). *Sas4* cKO spermatids either inherit only one mature centriole or no centriole. When one mature centriole is inherited, MT extension from the mature centriole, or basal body, is executed to form the flagella. However, the immature centriole is not present to form the centriole adjunct necessary for serving as the base for manchette MT nucleation and ensuring the attachment of the flagella to the sperm head. Therefore, flagella without a proper attachment to the head are lost. When *Sas4* cKO spermatids inherit no centrioles, they display similar defects as spermatids that inherit one mature centriole. In addition, because they lack the mature centriole they cannot form flagella. As a result, by steps 13-16, all *Sas4* cKO spermatids exhibit misshaped DNA, abnormal acrosome morphology, and lack flagella. Grey rectangle with rounded corners = mature centriole, green bar = maturation marker, yellow rectangle with rounded corners = immature centriole, yellow and purple oval = PCM and MT associated proteins, purple oval = microtubule-associated proteins, green oval = basal centriole, dark red oval = centriole adjunct, blue = DNA, pink = flagella, lime green = acrosome, yellow = manchette, red = CCDC13.

an important role during spermiogenesis by contributing PCM proteins to the spermatid neck and helping facilitate the head-tail attachment, establishing protein trafficking structures, and forming flagella. Spermatids that do not inherit both centrioles are unable to successfully execute these remodeling processes that are essential for successful spermatozoa formation. By the maturation phase, all *Sas4* cKO spermatids lack flagella and have experienced defects in protein trafficking that result in aberrant DNA and acrosome morphologies. Taken together, the defects in spermiogenesis exemplify the importance of inheriting two centrioles for head remodeling and flagella formation and attachment (Fig 8A and 8B).

## Discussion

### Meiotic divisions do not require centrioles

We have identified SAS4 as a critical centriole component required for centriole duplication in mammalian spermatocytes. Our assessment of *Sas4* cKO male mice has also indicated that despite centriole duplication failure, chromosome segregation occurs successfully during meiosis I and II, resulting in the formation of haploid spermatids that harbor either a single mature centriole or no centrioles at all (Fig 8A). The formation of haploid *Sas4* cKO spermatids relies on chromosome segregation being mediated by a ncMTOC during meiosis II. Based on the localization of MT-associated proteins CAMSAP1, TPX2, NuMA, and KIF11 to both centrosomal and ncMTOC spindle poles, it is likely these components play a critical role in facilitating bipolar spindle assembly during mammalian spermatogenesis. Prior *in vitro* and cell culture studies support this hypothesis [51]. For instance, CAMSAP1 and TPX2 have both been observed to promotes MT nucleation in the absence of the canonical MT nucleating structure, γ-TURC [69,71,72,74]. These proteins are also both MT stabilizing proteins that bind the polymerized MT structures and prevent catastrophe [41,70,72]. While CAMSAP1 binds primarily to MT minus ends, TPX2 binds more evenly along the entire length of the MTs [41,70,72]. NuMA is critical for spindle pole organization due to its ability to organize the MT minus to a uniform locus and coordinate bipolar spindle orientation with the cellular membrane [75,76,86]. KIF11 is a MT plus end direct molecular motor and is required for spindle pole separation [87–89]. The known roles of these MT-associated factors in MT nucleation, stabilization, organization, and separation into functional bipolar spindles in other cell types indicate they are likely critical to the ncMTOC observed in this study. However, the observations made in our study with spermatocytes

contrast previous reports that determined centriole duplication failure caused by *Sas4* mutation in somatic cells triggered a p53-dependent apoptotic pathway leading to subsequent cell death [45,46,90–92]. Overall, the two studies assessing *Sas4* and *Plk4* cKO demonstrate that spermatocytes do not require centrioles for meiotic division in the same way mitotically dividing cell types do [51]. Instead, a specific subset of centrosome-independent MT-associated proteins are sufficient for supporting chromosome segregation in primary and secondary spermatocytes.

## Centrioles are required for spermiogenesis

Despite executing meiotic divisions, *Sas4* cKO spermatids were unable to complete spermiogenesis successfully. Spermiogenesis is a necessary event for spermatozoa formation where major cellular remodeling takes place. One key aspect of spermiogenesis that centrioles are known to be required for is flagella formation. The centrioles serve as the base for flagella axoneme MT extension and contribute centriole and centrosome proteins necessary for proper head-tail attachment within the neck of the spermatozoa [4,29]. We observed the absence of flagella in *Sas4* cKO step 13–16 spermatids (Figs 7H, 7I, and 8B). We attribute this phenotype to both the failure of flagella formation in spermatids inheriting 0 centrioles, and the failure to maintain the head-tail attachment in spermatids that inherited only a single mature centriole. The immature centriole forms the centriole adjunct that is a structure necessary for the HTAC [3,29]. Without this immature centriole, the head-tail attachment is incomplete and connection of the flagella to the head of the spermatid cannot be maintained. Inheritance of at least one centriole is necessary for the spermatid to also inherit the associated centrosomal proteins, which have been demonstrated in several reports to be critical for the head-tail attachment. Mutations in centrosomal proteins such as *Cep112*, *Cep250*, *Cntrob*, and *Odf1* result in defects in head-tail attachment and can lead to headless sperm and non-obstructive azoospermia that contributes to infertility [30–35]. While mouse spermatozoa have been characterized to not have visible centriole structures by the time spermatozoa formation is complete, centriole and centrosome proteins get incorporated into flagella structures, making the inheritance of a full centrosome important for tail formation [3,93].

Additionally, failure to inherit two centrioles appears to cause defects in structures critical to head remodeling. The manchette and perinuclear ring are two structures within the spermatid head that help to define the polarity of the spermatid and serve as a protein trafficking network [94]. Perinuclear ring morphology and manchette formation and disassembly were observed to be defective between step 9 and 12 of spermiogenesis (Figs 7C-G, and 8B). These results suggest that there is a mis-regulation of the microtubule-based protein trafficking system leading to subsequent failure in the remodeling required for spermatozoa formation. This can be seen in the malformation of the DNA and the failure of acrosome cap development in elongating *Sas4* cKO spermatids (Figs 7A and 8B). Aberrant spermiogenesis resulting in a similar phenotype has been previously observed when the centrosomal gene *Cep78* is mutated, which resulted in low sperm count, impaired sperm motility, abnormal sperm morphology, and ultimately male infertility [95,96]. This condition known as OATS (oligoasthenoteratozoospermia) has been implicated as the cause of upwards of 30% of all male infertility cases [97]. It is likely the centrosomes play a role in MT organization and the formation of the transient manchette and perinuclear ring structures and defects in the process lead to downstream failure of acrosome formation and aberrant DNA morphology, contributing to infertility.

## Centriole duplication in other organisms

The mutation of *Sas4* has also been assessed for defects in other model organisms. SAS-4 in *Caenorhabditis elegans* is required for maintenance of centriole MT recruitment and stabilization, and its depletion leads to centriole duplication failure [98,99]. Partial depletion of SAS-4 leads to aberrant centrioles with less SAS-4 incorporated into the centriole structure and a lesser ability to recruit PCM factors leading to an overall smaller centrosome [99]. It should be noted, however, that the stability of SAS4 in the centriole structure varies across species. While *C. elegans* exhibit stably incorporated SAS-4 in their centrioles, human centrioles have been shown to incorporate CPAP (SAS4 ortholog) in a dynamic manner with the cytoplasmic pool of CPAP [100]. Additionally, human centrioles do not fail to recruit PCM to the centrioles when CPAP is depleted, which may, in part, be a result of the dynamic localization of PCM components [100].

Intriguingly, during spermatogenesis in *Drosophila melanogaster Sas-4* mutants resulted in multipolar spindles in primary spermatocytes and led to failure of meiotic divisions and abnormal spermatid elongation [101]. The multipolar spindle phenotype discovered when assessing *D. melanogaster Sas-4* mutants is in contrast with our observations in the *Sas4* cKO mouse spermatocytes that complete spermatogenesis despite centriole duplication failure (Figs 3, 6, and 8A). It is important to consider that canonically *D. melanogaster* spermatids complete spermiogenesis after inheriting only a single centriole, compared with mammalian spermatocytes that inherit two. However, despite this difference in centriole inheritance, the observation of abnormal spermiogenesis aligns with this study's findings. Additional corroboration of our results has been observed in other centriole gene knockout experiments in *D. melanogaster* where the depletion of centriolar proteins such as CETN1 results in male infertility [102]. This further exemplifies the importance of the centrioles in spermiogenesis across species. Taken together, work previously performed using non-mammalian models aligns with our observations of spermiogenesis in the mouse model assessed while also highlighting several key differences in centriole biogenesis between species that emphasize the importance of conducting mammalian assessments.

### Ciliopathies and infertility

There is increasing evidence demonstrating many ciliopathies first detected in non-reproductive tissues also lead to reproductive tissue disorders with symptoms of sub-fertility or infertility. Examples include conditions such as Bardet-Biedl, Kartagener, or Usher syndrome [5–10]. Furthermore, mutations in centrosomal proteins such as CEP78 have been implicated in similar multi-system ciliary perturbations with defects in fertility [103]. With advancing developments in therapies to treat individuals with ciliopathies, many individuals with these conditions desire to conceive children of their own. To navigate infertility, techniques such as traditional in-vitro fertilization (IVF) or intracytoplasmic sperm injection (ICSI) cannot be utilized due to the types of non-obstructive azoospermia ciliopathies typically cause [104–106]. ICSI can be employed utilizing spermatozoa that do not meet the typical motility requirements clinics have for sperm selection [104]. However, motility is not the only defect seen in some patients [104]. Emerging techniques such as the use of microdissection testicular sperm extraction (mTESE), round spermatid injection (ROSI), or elongating spermatid injection (ELSI) could be explored for these patients [106–109]. Continued assessments and understanding of the centrosome in the context of spermatogenesis are important to better advance these treatment options for use as assisted reproductive technologies. Another aspect that must be considered is the integrity of the genome within spermatids that have undergone abnormal elongation processes. The DNA fragmentation assays could be used to assess fecundity of the spermatids produced in patients with non-obstructive azoospermia ciliopathies [110,111].

### Conclusion

Successful completion of meiotic divisions and spermatid remodeling is required for spermatozoa formation. In this study, we have demonstrated that SAS4 is required for centriole duplication in spermatocytes. However, secondary spermatocytes can segregate their chromosomes without centrioles utilizing a ncMTOC mechanism. We have also identified that despite meiotic completion, two centrioles are required for spermiogenesis. Aberrant spermatid head remodeling, failure of flagella formation, and failure to maintain the head-tail attachment arise as a consequence of inheriting less than two centrioles. Future work assessing the role of centriole and centrosome components in spermatid remodeling are critical next steps to further understand the importance of the centrosome during spermiogenesis.

### Materials and methods

#### Ethics statement

All mice were bred at Johns Hopkins University (JHU, Baltimore, MD; MO19H08) and the Uniformed Services University of Health Sciences (USUHS, Bethesda, MD; BIO-22–090), and protocols for their care and use were approved by the

Institutional Animal Care and Use Committees (IACUC) of JHU and USUHS. All research was in accordance with the National Institutes of Health and U.S. Department of Agriculture guidelines.

**Mice.** Mice harboring the *Sas4* cKO allele (C57BL/6N; *Cenpi*tm1c(EUCOMM)Wtsi) used in this study have been previously reported [51]. Within the *Sas4* allele are two loxP sites flanking the 5th exon of *Sas4* (Fig 1A; [47]). Mice harboring the Centrin-GFP transgene (CB6-Tg(CAG-EGFP/CETN2)3–4Jgg/J) were obtained from the Jackson Laboratory [61]. Centriole duplication has been previously reported to occur normally in the spermatocytes of mice harboring the *Cetn2-Gfp* transgene [51]. Conditional mutation of *Sas4* was achieved using a hemizygous *Spo11* (Tg(Spo11-Cre)1Rsw/PecoJ) Cre recombinase transgene. The *Spo11* promoter was used to drive Cre recombinase expression in germ cells entering the meiotic program [48].

**PCR genotyping.** Primers used during this study are described in S1 Table. PCR conditions: 90°C for 2 min; 30 cycles of 90°C for 20 s; 58°C for 30 s; and 72°C for 1 min; and a final 10 min at 72°C.

**Tubule squash preparations.** Mouse tubule squash preparations were performed as previously described [112,113]. Primary and secondary antibodies and dilutions used are presented in S2 Table. Tubule squash preparations were mounted in Vectashield + DAPI (4', 6-diamidino-2-phenylindole) medium (Vector Laboratories). Full Z-stack captured images were utilized to manually identify centrioles, centrosomes, spindle morphology, and chromosome structure.

**Chromatin spread preparations.** Mouse chromatin spread preparations were performed as previously described [49]. Primary and secondary antibodies and dilutions used are presented in S2 Table. Chromatin spread preparations were immunolabeled and mounted in Vectashield + DAPI (Vector Laboratories). Full Z-stack captured images were utilized to manually identify chromosome structure.

**Histology and cryo-sectioning.** For histological assessment, mouse testis and epididymis tissue were fixed in bouins fixative (Ricca Chemical Company) before paraffin embedding. Serial sections 5 microns thick were mounted onto slides and stained with hematoxylin and eosin. To assess testis cryosections, testes were embedded in O.C.T. compound (Fisher) and frozen at -80°C. Serial sections, 16 microns thick, were mounted onto slides and immunolabeled with primary and secondary antibodies (S2 Table).

**Epididymal sperm count.** Mouse epididymides (one per mouse) were dissected and placed into PBS, then cut into several smaller pieces and incubated for 30 minutes at 30°C to allow time for sperm to be released from the tissue. The sperm-containing solution was then transferred to a fresh tube, separating released sperm from epididymal tissue, and diluted as necessary before counting sperm with a hemocytometer.

**Microscope image acquisition.** Tubule squash and chromatin spread images were captured using a Zeiss CellObserver Z1 linked to an ORCA-Flash 4.0 CMOS camera (Hamamatsu). Images were processed and analyzed using ZEN 2012 blue edition imaging software. Photoshop (Adobe) was used to prepare the figure images.

Histology and immunohistology of sectioned testes were captured using a Keyence BZ-X800 and analyzed using the BZ-X800 Viewer and Analyzer software. (Adobe) was used to prepare the figure images.

**Microtubule length and width measurements.** Spindle measurements were made using ZEN 2012 blue edition imaging software (Zeiss). The spindle length was obtained by measuring from where microtubule nucleation began at an MTOC to the aligned chromosomes on the metaphase II spindle. The width and length of both spindles of a metaphase II spermatocyte were measured and assessed as a ratio of A:B, with A being the smaller of the two length or width measurements, in line with similar measurements in a previous report [51].

**Data and statistical analysis.** All graphs and statistical analyses were performed with GraphPad Prism 10.4.1 software. Details of cell and animal counts and statistical analyses are provided in each figure legend.

## Supporting information

**S1 Table. Genotyping primers used in this study.**
(XLSX)

**S2 Table. Antibodies used in this study.**
(XLSX)

**S1 Data. Raw data for all quantified values in this study.**
(XLSX)

## Acknowledgments

The authors thank Dr. Hisham Bazzi for *Sas4* flox/flox mice (*Cenpj*<sup>tm1c(EUCOMM)Wtsi</sup>). We would also like to thank Tara Little, Chris Shults, and Jingwen Xu for their pilot studies. The opinions and assertions expressed herein are those of the author(s) and do not reflect the official policy or position of the Uniformed Services University of the Health Sciences or the Department of Defense.

## Author contributions

**Conceptualization:** Marnie W. Skinner, Philip W. Jordan.

**Data curation:** Marnie W. Skinner, Paula B Nhan, Carter J. Simington, Philip W. Jordan.

**Formal analysis:** Marnie W. Skinner, Paula B Nhan, Carter J. Simington, Philip W. Jordan.

**Funding acquisition:** Marnie W. Skinner, Philip W. Jordan.

**Investigation:** Marnie W. Skinner, Carter J. Simington, Philip W. Jordan.

**Methodology:** Marnie W. Skinner, Philip W. Jordan.

**Project administration:** Philip W. Jordan.

**Resources:** Philip W. Jordan.

**Supervision:** Philip W. Jordan.

**Validation:** Marnie W. Skinner, Paula B Nhan, Philip W. Jordan.

**Visualization:** Marnie W. Skinner, Paula B Nhan, Philip W. Jordan.

**Writing – original draft:** Marnie W. Skinner, Philip W. Jordan.

**Writing – review & editing:** Marnie W. Skinner, Paula B Nhan, Carter J. Simington, Philip W. Jordan.

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
