## [Decision Letter · Decision Letter 0]

24 Mar 2025

PGENETICS-D-25-00228

Meiotic divisions and round spermatid formation do not require centriole duplication in mice

PLOS Genetics

Dear Phil,

Thank you for submitting your manuscript to PLOS Genetics. After careful consideration, we feel that it has merit but does not fully meet PLOS Genetics's publication criteria as it currently stands. Therefore, we invite you to submit a revised version of the manuscript that addresses the points raised during the review process.

Please submit your revised manuscript within 30 days Apr 23 2025 11:59PM. If you will need more time than this to complete your revisions, please reply to this message or contact the journal office at plosgenetics@plos.org. Please include the following items when submitting your revised manuscript:

We look forward to receiving your revised manuscript.

Kind regards,

Paula E. Cohen

Section Editor

PLOS Genetics

Aimée Dudley

Editor-in-Chief

PLOS Genetics

Anne Goriely

Editor-in-Chief

PLOS Genetics

**Journal Requirements:**

1) Please upload all main figures as separate Figure files in .tif or .eps format. For more information about how to convert and format your figure files please see our guidelines: 

2) We have noticed that you have uploaded Supporting Information files, but you have not included a list of legends. Please add a full list of legends for your Supporting Information files after the references list.

3) Some material included in your submission may be copyrighted. According to PLOSu2019s copyright policy, authors who use figures or other material (e.g., graphics, clipart, maps) from another author or copyright holder must demonstrate or obtain permission to publish this material under the Creative Commons Attribution 4.0 International (CC BY 4.0) License used by PLOS journals. Please closely review the details of PLOSu2019s copyright requirements here: PLOS Licenses and Copyright. If you need to request permissions from a copyright holder, you may use PLOS's Copyright Content Permission form.

Potential Copyright Issues:

i) Figures 3F, 4A, 5D, and 8. Please confirm whether you drew the images / clip-art within the figure panels by hand. If you did not draw the images, please provide (a) a link to the source of the images or icons and their license / terms of use; or (b) written permission from the copyright holder to publish the images or icons under our CC BY 4.0 license. Alternatively, you may replace the images with open source alternatives. See these open source resources you may use to replace images / clip-art:

4) Please amend your detailed Financial Disclosure statement. This is published with the article. It must therefore be completed in full sentences and contain the exact wording you wish to be published.

2) If any authors received a salary from any of your funders, please state which authors and which funders.

5) We note that the Data Availability Statement mentioned in the manuscript is different from that provided in the online submission form. The Data Availability statement in the online submission form is currently as follows: 'All data underlying the findings presented in this manuscript will be made fully available, without restriction, upon request.' While the one in the manuscript states 'All raw data underlying the findings reported within this manuscript are available at:

http://datadryad.org/stash/share/BD1LjvEVFKYEC_CiTiKaQ0uR8JoofTF_EFqSBVVUtyE.' Please provide the complete Data Availability statement in the submission form and ensure that it matches the one mentioned in the manuscript.

**Reviewers' comments:**

Reviewer's Responses to Questions

Reviewer #1: The manuscript by Skinner et. al. describes the effects of a sas4 conditional knockout on meiotic chromosome segregation during spermatogenesis and spermiogenesis. Sas4 conditional knockout male germ cells exhibit a failure to duplicate the centrioles in both meiosis I and meiosis II. Interestingly, despite a high percentage of meiosis II spermatocytes only having one centriole, these germ cells are still capable of forming a bipolar spindle and segregating chromosomes. The authors found that the acentriolar MTOCs contain MT-associated proteins. Even though chromosome segregation occurs in the absence of centriole duplication, the authors demonstrate that Sas4 and centriole duplication are required for accurate spermiogenesis.

Major comments:

1. Figure 3E. The authors show that centriole duplication fails in both meiosis I and meiosis II with ~60% of meiosis II spermatocytes with only 1 centriole per cell. The images show localization of both centrioles (CETN3) and spindle poles (alpha-tubulin). The representative image with only one centriole looks like it is still capable of forming a bipolar spindle. Is a bipolar spindle observed in all cells with only 1 centriole? Are monopolar spindles ever observed?

Related to these questions, it would be helpful to bring number of spindles and whether they are centriolar or acentriolar at the end of the paragraph on page 12 (~lines 205-213).

2. On page 13 lines 244-245, the authors state that they wanted to examine whether oocyte acentriolar PCM proteins were present in the sas4 cKO acentriolar pole, yet both CEP192 and GCP2 are found in the centriolar based poles in control spermatogenesis. Either this sentence needs to be reworded (maybe to say lacking canonical PCM proteins) or oocyte specific PCM proteins needs to be used for the experiments.

3. Figure 3D shows that a higher percentage of metaphase II cells have 1 centriole, which I presume would lead to one spermatid with 1 centriole and one with 0. But there are also ~20% with 2 centrioles, which should give rise to two spermatids each with 1 centriole. Can the authors speculate on why there is a higher percentage of round spermatids and elongating spermatids with 0 centrioles rather than 1 centriole (Figure 6B & 6C)?

4. In the discussion (page 18, lines 372-373), the authors state that a subset of MT-associated proteins are sufficient for chromosome segregation. This conclusion seems a little strong as sufficiency was not tested.

Minor suggestions:

1. The yellow numbers indicating centriole numbers in figures 3 and 4 are hard to see. White numbers might provide more contrast with the black background.

2. Figure 4E figure legend- need to define ncMTOC as this isn’t defined in the text until discussion of Figure 5.

3. Define aMTOC on page 13 line 242.

4. Very minor- Line 265 on page 14 change “an” to “a”. Should say which is a MT motor protein

Reviewer #2: This is a well-written, clear and rigorous analysis of the defects observed when SAS4, a protein required for centriole biogenesis, is conditionally knocked out during spermatogenesis. In particular, the diagrams provided in the figures to orient the reader were very helpful. The authors report no defects in meiotic prophase but clear defects in centriole duplication. The defects in centriole duplication do not affect chromosome segregation in meiosis I or II because of the formation of noncentrosomal MTOCs but do affect later events in spermiogenesis, explaining the infertility defects they observe. I appreciated Figure 8, where the authors distinguished between the phenotypes observed in products of meiosis that did and did not receive a centriole. I have a few minor comments that might make this manuscript more widely accessible.

Line 155: “The large lumen diameter was due to the Sas4 cKO elongated spermatids not harboring flagella (Fig 1D).” Can the authors indicate in the control where the flagella are?

Can the authors provide an image of 4 centrioles in SAS4 cKO? Are there any differences observed between this class in control and SAS4 cKO?

I can’t see the maturation of the other parent centriole in the both control and cKO images in Figure 4C. I am much more convinced by the images in metaphase I. Would the pachynema images benefit from gray scale versions being available?

Can the authors provide more guidance in Figure 7 so that a reader unfamiliar with spermiogenesis can follow the phenotypes of the SAS4 cKO, especially in Figure 7E?

Reviewer #3: This manuscript by Skinner et al. focuses on the role of SAS4, a centriolar protein, during spermiogenesis. They make a conditional knockout of Sas4 and study the consequences during meiosis and the formation of spermatids. They find that spermatozoa are not functional, causing infertility. Upon further analysis, meiotic prophase events seem normal, except that centrioles fail to duplicate during zygonema. Meiotic events still occur, but centrioles fail to duplicate again for meiosis II. They also lack centrosomal proteins, making them non-centrosomal MTOCs. They do have MT-associated proteins to coordinate spindle assembly. Despite the centriolar failure, the spermatids have chromatin (suggesting meiosis I and II occurred), but the major problems arise in formation of normal sperm. Of the steps of spermiogenesis, the defects start in step 9 with abnormal acrosome width, manchette length, perinuclear ring diameter, and the lack of a flagella.

This manuscript has interesting findings, that will contribute to the role of centrioles in spermiogenesis. While the results are primarily descriptive, they provide a foundation for future work. There are some issues that need to be considered:

1) The conditional knockout was made with the expression of CRE under the Spo11 promoter, which is turned on in preleptotene/ leptotene stage. There is no check that there is no protein production and that the protein made prior to Spo11 induction of CRE is turned over rapidly. The experiments in figure 2 are looking at events that occur early in prophase, leptotene and early zygotene. They do not observe defects with the cKO and interpret the results as Sas4 not having a role in those events, unlike Cep63. However, there are no experiments showing that the Sas4 protein is no longer present during those stages. The authors should determine if the protein is no longer present by performing immunostaining, or some other assay to look at levels of the Sas4 protein.

2) The authors mention that meiosis II was successfully completed, but did they track whether chromosome segregation occured normally?

3) In Fig 7, multiple steps are represented in one image (eg. steps 1-7). Is this because you cannot tell the difference between steps 1-7? If you can tell the difference, it may be better to state the actual step imaged instead of lumping them all together.

4) In Fig 7H-I, what is the inset box in the left corner showing? The figure legends says DAPI, but is it the entire structure? The other inset boxes zoom into smaller structures.

**Have all data underlying the figures and results presented in the manuscript been provided?**

Reviewer #1: Yes

Reviewer #2: **No: ** The authors indicate that it will be provided upon request.

Reviewer #3: **No: ** numerical data underlying graphs was not provided

PLOS authors have the option to publish the peer review history of their article (what does this mean? ). If published, this will include your full peer review and any attached files.

**Do you want your identity to be public for this peer review?** For information about this choice, including consent withdrawal, please see our Privacy Policy .

Reviewer #1: No

Reviewer #2: No

Reviewer #3: No

**Figure resubmission:**
---

## [Editor Report · Decision Letter 1]

21 Apr 2025

Dear Phil,

We are pleased to inform you that your manuscript entitled "Meiotic divisions and round spermatid formation do not require centriole duplication in mice" has been editorially accepted for publication in PLOS Genetics. Congratulations!

Before your submission can be formally accepted and sent to production you will need to complete our formatting changes, which you will receive in a follow up email. In addition to these formatting changes, as a personal recommendation, we would suggest that you include in your discussion a brief explanation of why you could not demonstrate depletion of SAS4 - it is important to acknowledge this as a caveat, but also to emphasize (as you did in your rebuttal) why the biology indicates appropriate SAS4 depletion even if the antibodies failed.

Please be aware that it may take several days for you to receive your follow-up formatting email; during this time no action is required by you. Please note: the accept date on your published article will reflect the date of this provisional acceptance, but your manuscript will not be scheduled for publication until the required changes have been made. 

Yours sincerely,

Paula E. Cohen

Section Editor

PLOS Genetics

Paula Cohen

Section Editor

PLOS Genetics

Aimée Dudley

Editor-in-Chief

PLOS Genetics

Anne Goriely

Editor-in-Chief

PLOS Genetics

Comments from the reviewers (if applicable):

**Data Deposition**

http://datadryad.org/submit?journalID=pgenetics&manu=PGENETICS-D-25-00228R1

**Press Queries**

---

## [Editor Report · Acceptance letter]

PGENETICS-D-25-00228R1

Meiotic divisions and round spermatid formation do not require centriole duplication in mice

Dear Dr Jordan,

We are pleased to inform you that your manuscript entitled "Meiotic divisions and round spermatid formation do not require centriole duplication in mice" has been formally accepted for publication in PLOS Genetics! Your manuscript is now with our production department and you will be notified of the publication date in due course.

With kind regards,

Anita Estes

PLOS Genetics

On behalf of:
